# The Spectral Bias of Polynomial Neural Networks

**Moulik Choraria**
University of Illinois at Urbana-Champaign
moulikc2@illinois.edu

**Leello Dadi**
EPFL, Switzerland
leello.dadi@epfl.ch

**Grigorios G Chrysos**
EPFL, Switzerland
grigorios.chrysos@epfl.ch

**Julien Mairal**
Univ. Grenoble-Alpes, Inria
julien.mairal@inria.fr

**Volkan Cevher**
EPFL, Switzerland
volkan.cevher@epfl.ch

## Abstract

Polynomial neural networks (PNNs) have been recently shown to be particularly effective at image generation and face recognition, where high-frequency information is critical. Previous studies have revealed that neural networks demonstrate a *spectral bias* towards low-frequency functions, which yields faster learning of low-frequency components during training. Inspired by such studies, we conduct a spectral analysis of the Neural Tangent Kernel (NTK) of PNNs. We find that the Π-Net family, i.e., a recently proposed parametrization of PNNs, speeds up the learning of the higher frequencies. We verify the theoretical bias through extensive experiments. We expect our analysis to provide novel insights into designing architectures and learning frameworks by incorporating multiplicative interactions via polynomials.

## 1 Introduction

Deep neural networks (DNNs) have demonstrated remarkable success in different domains [22, 11]. DNNs can approximate complex functions or even datasets with randomized labels arbitrarily well [50, 3], which makes their ability to avoid over-fitting on real data surprising, since it seems to disagree with prior notions of model complexity. This has sparked the interest in investigating the notion of "implicit bias" in neural network training, which makes them favor low complexity solutions [43, 21, 26].

The spectral analysis of deep networks offers one perspective on this implicit bias. Deep neural networks demonstrate a learning bias towards low frequency functions - i.e. functions that vary globally without local fluctuations are learned faster when training neural networks via gradient descent [37, 48]. The phenomenon, termed as the *spectral bias* of neural networks [37], has been explored from the perspective of the Neural Tangent Kernel (NTK) [24]. The eigenvalues of the obtained kernel influence important characteristics such as the approximation properties and rate of learning [10, 39, 36]. For standard two-layer ReLU networks within the NTK regime, the analysis supports the idea of a spectral bias by showing faster error convergence for information in lower frequencies [12].

Recently, another class of models, Polynomial Neural Networks (PNNs), that express high order polynomial expansions, have demonstrated state-of-the-art performance in the challenging tasks of image generation [28] and face recognition [17]. Both tasks rely on fine-grained details, which correspond to high-frequency information. More generally, multiplicative interactions have demonstrated strong empirical performance on image-based applications [44, 47, 4]. Jayakumar et al. [25] highlight how multiplicative interactions, which construct second degree polynomials, can enlarge the hypothesis space and lead to faster learning for certain classes of functions.

To understand this success of PNNs, we conduct a spectral analysis, inspired by the analysis of DNNs. We focus on one instance of polynomial networks called $\Pi$-Nets [16], where the output is a piece-wise polynomial function of the input obtained via multiplicative layers. The parameters of a $\Pi$-Net can be represented as high-order tensors, while polynomial expansions can offer increased representation power [17]. Our main contributions can be summarized as follows:

1. We analyze two-layer polynomial networks in the NTK regime. By studying the spectral properties of the corresponding kernel, we prove a theoretical speed-up in learning higher frequencies over standard neural networks and validate the hypothesis in the approximate NTK regime, on the task of learning spherical harmonics.

2. Beyond the NTK regime, we demonstrate this enhanced bias of $\Pi$-Nets towards higher frequencies in several experimental settings, beginning from synthetic learning tasks and then proceeding to state-of-art networks and inverse problems with 2D images.

Aside from improving the understanding of polynomials in neural networks, our proposed analysis also sheds new light on the effect of multiplicative interactions in neural networks, prevalent in certain domains of machine learning including vision and natural language processing [29, 7].

## 2 RELATED WORK

**Spectral Bias**: Motivated by the empirical observations in [3, 37, 48, 46] that deep networks first learn "simple patterns", several papers [49, 12, 8, 2] have conducted theoretical analyses to explain this bias towards lower frequencies. The work of [2], aiming to understand why random labels take longer to learn than natural labels, showed that alignment of the labels with the eigenvectors of the NTK Gram matrix determines the learning speed. Extending this result, Cao et al. [12] provide an explanation for the spectral bias by analyzing the decay rate of eigenvalues of the NTK when the input data is uniformly distributed on the sphere. Under the same assumption, Basri et al. [8] study training dynamics in the NTK setting for 2-layer ReLU networks with a fixed outer layer and an explicitly included linear bias term in the ReLU. [9] further extends this work to account for non-uniform data distributions. All these findings show that DNNs learn lower frequency functions faster which prompted Tancik et al. [45] to propose methods to mitigate this bias. Our paper aims to establish similar results for PNNs, to explain their good performance at learning higher frequencies.

**Polynomial neural networks (PNNs)**: The early papers that explore polynomials in the context of neural networks are mainly divided into two categories: 1) self-organizing networks with hard coded features [23], 2) Pi-Sigma networks [41, 34]. In both cases, the constructions did not scale well to higher dimensional inputs, and were not used for high-dimensional signals, such as images. More recent papers have used the Hadamard product to capture correlations between different branches of an architecture [4, 44, 47, 17]. Our goal is to analyze the properties of these polynomial neural networks that have shown sucess in practice.

**Polynomial activation functions (PAFs)**: It is important to note the distinction between PNNs and Polynomial activation functions. PAFs expand (element-wise) each feature to an $r^{\text{th}}$ degree, i.e., they assume a (deep) neural network where the element-wise activation functions are $r^{\text{th}}$ degree polynomials. This is substantially different from capturing higher-order correlations across input (or feature) elements like PNNs especially in the presence of non-linear activations. Theoretical work on over-parametrization [18], expressive power [30] and generalization of shallow nets [35] have emerged for PAFs. The aforementioned papers, however, do not conduct a spectral analysis and do not exhibit the benefits of PNNs for learning high-frequency information.

## 3 ANALYSIS OF POLYNOMIAL NEURAL NETWORKS IN THE NTK REGIME

In this section, we conduct a careful analysis of the kernel approximation of polynomial neural networks (PNNs) to gain insight on the effect of the multiplicative interactions. PNNs include multiplicative interactions and express high-degree polynomial expansions. The recent parametrization of the $\Pi$-Net family [16], which we summarize below, is used as a representative PNN. We review the tangent kernel approximation of neural networks and then we derive the tangent kernel of two-layer $\Pi$-Nets to study the spectral bias of $\Pi$-Nets.

**Notation**: We denote by $\langle \cdot, \cdot \rangle$ the standard inner-product on $\mathbb{R}^d$. For two vectors $\boldsymbol{x}, \boldsymbol{y} \in \mathbb{R}^d$, $\boldsymbol{x} * \boldsymbol{y}$ denotes the element-wise or Hadamard product. We use $\times_m$ to denote the mode-$m$ vector product[1].

---

[1]The reader may refer to the appendix for more details on the mode-m product.

We define the asymptotic notations $\Omega(\cdot)$ and $\tilde{\Omega}(\cdot)$ as follows: Let $a_n$ and $b_n$ be two sequences. We write $a_n = \Omega(b_n)$ if $\liminf_{n\to\infty} |a_n/b_n| > 0$. We use $\tilde{\Omega}(\cdot)$ to hide the logarithmic factors in $\Omega(\cdot)$.

## 3.1 Π-NET FORMULATION

A polynomial expansion of the input vector $z \in \mathbb{R}^\delta$ can be used to express the output $x \in \mathbb{R}^o$ as an $N^{\text{th}}$ degree polynomial expansion as follows:

$$x = \sum_{n=1}^{N} \left( \mathcal{W}^{[n]} \prod_{j=2}^{n+1} \times_j z \right) + \beta, \tag{1}$$

where $\beta \in \mathbb{R}^o$ and $\left\{ \mathcal{W}^{[n]} \in \mathbb{R}^{o \times \prod_{m=1}^{n} \times_m d} \right\}_{n=1}^{N}$ are the learnable parameters, and $\times_m$ is the mode-$m$ vector product. Since the number of tensor parameters $\mathcal{W}^{[n]}$ grow exponentially with the degree of the polynomial, a coupled tensor decomposition with factor sharing can be used. The idea in Π-Nets is to propose such decompositions that can capture higher order correlations, and can be implemented in standard deep learning frameworks. One such decomposition uses the recursive formulation $x_n = \left( A_{[n]}^T z \right) * \left( S_{[n]}^T x_{n-1} + B_{[n]}^T b_{[n]} \right)$ for $n = 1, \ldots, N$ and expresses the output $x$ as $x = C x_N + \beta$. The term in the rightmost parenthesis is exactly the recursive form of a standard neural network (without activations). Therefore, Π-Nets augment standard neural network with multiplicative interactions via the Hadamard product. While Π-Nets can approximate the target function without activations, *they achieve state-of-art performance with activation functions*, wherein the output is a piece-wise polynomial. We include more details in the Appendix.

## 3.2 THE NEURAL TANGENT KERNEL

Consider the following two-layer ReLU neural network (without bias parameters) with width $m$ that assumes the following form (defined as in [10]): $f_W(x) = \sqrt{\frac{2}{m}} W_2 \sigma(W_1 x)$, where $W_1 \in \mathbb{R}^{m \times (d+1)}$, $W_2 \in \mathbb{R}^{1 \times m}$ and we assume inputs $\{x\}_{i=1}^n$ follow some distribution $\tau$ on the unit sphere $\mathbb{S}^d \in \mathbb{R}^{d+1}$; $\sigma(\cdot)$ denotes the element-wise ReLU operator. As the width of the network $m$ goes to infinity, if the weights at initialization $W^{(0)}$ are independent and each follow the standard normal distribution, the inner product of the network gradient at initialization gives rise to a limiting kernel, namely the *Neural Tangent Kernel* (NTK) [24] $\kappa$ defined as :

$$\kappa(x, x') = \lim_{m\to\infty} \langle \nabla_W f_{W^{(0)}}(x), \nabla_W f_{W^{(0)}}(x') \rangle. \tag{2}$$

This kernel has been used to characterize the behavior of sufficiently wide networks $f_W$ during training. For instance, if the network is trained to minimize the $\ell_2$ loss, then its training dynamics closely track those of kernel regression under $\kappa$. This holds in a particular training regime, referred to as "lazy training" [14], where the parameters hardly vary after initialization and the network can be approximated by its first-order Taylor expansion at initialization as:

$$f_W(x) \approx f_{W^{(0)}}(x) + \langle \nabla_W f_{W^{(0)}}(x), W - W^{(0)} \rangle.$$

In practice however, the conditions of lazy training are often violated (notably, the requirement that the weights do not move) within the first few steps of gradient descent. Nevertheless, the NTK remains a useful theoretical tool for analyzing the neural network behavior as some of its predictions have been shown to hold in practice [12, 32].

The NTK for the two-layer ReLU network $f_W(x)$ takes the following form [10, 14, 19]

$$\kappa(x, x') = 2\langle x, x' \rangle \kappa_1(x, x') + 2\kappa_2(x, x'), \tag{3}$$

where the kernels $\kappa_1$ and $\kappa_2$ are defined as follows:

$$\kappa_1(x, x') = \mathbb{E}_{w \sim N(0, I)}[\sigma'(\langle w, x \rangle)\sigma'(\langle w, x' \rangle)],$$
$$\kappa_2(x, x') = \mathbb{E}_{w \sim N(0, I)}[\sigma(\langle w, x \rangle)\sigma(\langle w, x' \rangle)], \tag{4}$$

where $\sigma(\cdot)$ denotes the ReLU operator and $\sigma'(\cdot)$ denotes the indicator function $\mathbf{1}[\cdot \geqslant 0]$. The kernels $\kappa_1, \kappa_2$ have been explicitly computed for the ReLU activation in [38, 15, 10] where they were shown to be positive semi-definite *dot-product* kernels. The function value depends only on the value of the dot-product of the arguments or more formally, $\kappa_1(x, x') = g_1(\langle x, x' \rangle)$ and $\kappa_2(x, x') = g_2(\langle x, x' \rangle)$ for some functions $g_1, g_2 : \mathbb{R} \to \mathbb{R}$. These kernels will serve as building blocks for the tangent kernel of the Π-Net architecture studied in the next section.

### 3.3 Π-KERNEL

Following [8, 10, 12] that consider shallow two-layer ReLU networks, we derive the tangent kernel for a two-layer Π-Net by supplementing the standard network with a multiplicative interaction layer:

$$f_{\boldsymbol{W}}(\boldsymbol{x}) = \sqrt{\frac{2}{m}} \boldsymbol{W_3}[\sigma(\boldsymbol{W_2 x}) * \sigma(\boldsymbol{W_1 x})], \tag{5}$$

where $\boldsymbol{W_1}, \boldsymbol{W_2} \in \mathbb{R}^{m \times (d+1)}$, $\boldsymbol{W_3} \in \mathbb{R}^{1 \times m}$, $*$ denotes the Hadamard product and we again assume inputs $\{\boldsymbol{x}\}_{i=1}^n$ that follow some distribution $\tau$ on the unit sphere $\mathbb{S}^d \subset \mathbb{R}^{d+1}$.

**Remark 1** Note that formulation yields a piece-wise quadratic polynomial and that it is important to consider ReLU activations (or other non-linearities) since otherwise, the network yields a quadratic polynomial of the input and is no longer a universal approximator, even with infinite width.

**Theorem 1** *Let $\kappa_1, \kappa_2$ as defined in Eq. (4). The limiting kernel for the two-layer Π-Net, called Π-kernel and denoted by $\boldsymbol{\kappa_\pi}(\boldsymbol{x}, \boldsymbol{x'})$, takes the following form:*

$$\boldsymbol{\kappa_\pi}(\boldsymbol{x}, \boldsymbol{x'}) = 2(2\langle \boldsymbol{x}, \boldsymbol{x'} \rangle \boldsymbol{\kappa_1}(\boldsymbol{x}, \boldsymbol{x'}) + \boldsymbol{\kappa_2}(\boldsymbol{x}, \boldsymbol{x'}))\boldsymbol{\kappa_2}(\boldsymbol{x}, \boldsymbol{x'}). \tag{6}$$

The proof follows the standard NTK calculations and is presented in the Appendix. Importantly, the addition of a single multiplicative interaction induces a product of kernels form.

**Remark 2** The required width that ensures that the two-layer Π-Net stays close to initialization (the corresponding NTK is close to the limiting $\boldsymbol{\kappa_\pi}$) is slightly higher than standard networks, by a $\Omega(\sqrt{\log m})$ factor. We provide a rough sketch of the proof in the Appendix (end of C).

**Remark 3** The advantage of using the 2-layer Π-Net formulation is that it allows us to directly contrast against prior results on standard 2-layer feed-forward network, by gauging the effect of the extra multiplicative layer. However, unlike feed-forward networks, the theory does not easily extend to polynomials of higher degree or depth, since even with a fixed degree and depth, the NTK varies with the placement of the multiplicative connections. In this work, we choose to focus primarily on the effect of the multiplicative layer and leave the extension to general polynomials for future work.

Since the Π-kernel $\boldsymbol{\kappa_\pi}$ can be expressed as a sum of products of continuous positive definite dot-product kernels $\kappa_1$ and $\kappa_2$, it inherits their regularity properties and is itself a Mercer kernel, as a consequence of the Schur Product theorem.

### 3.4 SPECTRAL ANALYSIS

To properly characterize the approximation properties and the behavior during training of this newly derived kernel, we study its Mercer decomposition. Indeed, this approach was used in [10] to show that the 2-layer NTK (Eq. (3)) has better approximation properties than a fixed first layer network, and in [12] to study the spectral bias of feed forward ReLU networks.

The Mercer decomposition of a kernel $\kappa$ is derived from the eigenvalues and eigenfunction of the integral operator associated to the kernel (see [40] Theorem 2.10 and references therein). Taking $\mathcal{X}$ to be some compact set in $\mathbb{R}^d$, recall that for any continuous kernel function $\kappa : \mathcal{X} \times \mathcal{X} \to \mathbb{R}$ and Borel measure $\tau$ on $\mathcal{X}$, we can define an integral operator $L_\kappa$ that "convolves" any square integrable function $f \in L_\tau^2(\mathcal{X})$ with $\kappa$:

$$L_\kappa(f)(\boldsymbol{x}) = \int_X \kappa(\boldsymbol{x}, \boldsymbol{y}) f(\boldsymbol{y}) d\tau(\boldsymbol{y}). \tag{7}$$

For a Mercer kernel, this linear operator admits countable, real, non-negative eigenvalues $\{\mu_1, \mu_2, \dots\}$ and the associated eigenfunctions form an orthonormal basis of $L_\tau^2(\mathcal{X})$. If the data is uniform on the sphere $\mathbb{S}^d$, and $\kappa$ is a dot-product kernel, then these eigenfunctions of $L_\kappa$ are the spherical harmonics ([42], Lemma 4). Consequently, by applying Mercer's Theorem, we can obtain the following decomposition:

$$\kappa(\boldsymbol{x}, \boldsymbol{x'}) = \sum_{k=0}^{\infty} \mu_k \sum_{j=1}^{N(d,k)} Y_{k,j}(\boldsymbol{x}) Y_{k,j}(\boldsymbol{x'}), \tag{8}$$

where $Y_{k,j}$ for $j = 1, 2 \dots N(d, k)$ represent the spherical harmonics of degree $k$ in $d + 1$ variables (whose explicit formula is given in Appendix D.1 in [5]) , and $N(d, k) := \frac{2k+d-1}{k} \binom{k+d-2}{d-1}$.

From this, we can characterize the RKHS $\mathcal{H}$ associated to $\kappa$ as follows:

$$\mathcal{H} = \left\{ f = \sum_{k \geqslant 0, \mu_k \neq 0} \sum_{j=1}^{N(d,k)} a_{k,j} Y_{k,j}(\cdot) \text{ subject to } \|f\|_{\mathcal{H}}^2 = \sum_{k \geqslant 0, \mu_k \neq 0} \sum_{j=1}^{N(d,k)} \frac{a_{k,j}^2}{\mu_k} < \infty \right\}.$$

The benefit of determining the decay rate of the eigenvalues $\{\mu_1, \mu_2, \dots\}$ can be easily deduced from the previous set as the eigenvalues determine the *size* of $\mathcal{H}$. Indeed, the slower the decay of $\{\mu_1, \mu_2, \dots\}$, the more sequences $(a_{k,j})_{k,j}$ will verify $\sum_{k \geqslant 0, \mu_k \neq 0} \sum_{j=1}^{N(d,k)} \frac{a_{k,j}^2}{\mu_k} < \infty$. The results of [12] rely on this decay rate to show that gradient descent on wide networks picks up low-frequency information first[2] (Theorem 4.2, [12]). This decay rate is therefore of central importance. Next, we briefly refer to the prior results on characterizing this decay rate for the kernel obtained for the two-layer feed forward ReLU network (Eq. (3)) and we derive the equivalent results for the $\Pi$-kernel.

**Proposition 1** *(Theorem 4.3 in Cao et al. [12], Proposition 5 in Bietti & Mairal [10]) For the tangent kernel corresponding to two-layer feed forward ReLU network, the eigenvalues $(\mu_k)_k$ satisfy the following:*

$$\begin{cases} \mu_0, \mu_1 = \Omega(1), \\ \mu_k = 0, when\ k\ is\ odd, \\ \mu_k = \Omega(k^{-d-1})\ when\ k \gg d. \end{cases} \tag{9}$$

The decay is exponentially fast in the input dimension $d$. For the $\Pi$-kernel, we show the following improvement.

**Theorem 2** *Let $\{\mu_{\pi,1}, \mu_{\pi,2}, \dots\}$ denote the eigenvalues of the linear operator $L_{\kappa_\pi}$ associated to the kernel $\kappa_\pi$. For $k \gg d$ $(k \neq 2\ mod\ 4)$, it holds that $\mu_{\pi,k} = \Omega(k^{-d/2-2})$.*

The proof of the theorem is provided in the Appendix. The main idea is to plug in the Mercer decomposition of each kernel, leading to a product of polynomials form. The expansion of this product then allows for isolating the dominant terms contributing to the eigenvalues for each frequency. The key take-away is the much slower rate of decay the $k^{\text{th}}$ eigenvalue (an order almost $\Omega(k^{(d/2)})$) when compared to the standard two-layer NTK. An immediate consequence is a "larger" RKHS, which lends the $\Pi$-kernel superior approximation properties in the higher frequencies. Further, when combined with the findings of [12], it yields a speed-up in learning higher frequency harmonics.

**Remark 4** On the point of a "larger" RKHS, we note the two perspectives at play. First is the idea that a slower decay implies more functions contained in the RKHS, leading to better approximation properties. However, from the classical statistical learning point of view, while a larger RKHS makes the optimization problem easier, it may lead to a sub-optimal prediction performance [6], eventually relating to the traditional trade-off in machine learning. We note through our experiments that it is the perspective relating to better approximation and optimization for the $\Pi$-kernel that dominates.

**Remark 5** The $k \gg d$ setting may not reflect the case for image-based applications. For instance, generation tasks wherein the output is an image of resolution $d \times d$, the higher frequencies of interest roughly correspond to the order of $\Omega(d)$. Consequently, the exponential improvement in the decay rate for the $k \gg d$ setting, and by extension in the approximation properties of the RKHS as well as the speed of learning higher frequency information, may not be as dramatic elsewhere. Nevertheless, our analysis provides sufficient intuition to expect some degree of improvement in realistic settings.

## 4 NUMERICAL EVIDENCE

The analysis in Section 3 reveals that polynomial networks in the NTK regime learn higher frequency information faster. In practice however, neural networks deviate from the near-initialization NTK conditions within just a few iterations of gradient descent. Therefore, to verify the analysis on the spectral bias of $\Pi$-Nets, we conduct a series of experiments that increasingly deviate from the NTK regime, including image-based datasets to further verify our theoretical analysis. We initially consider synthetic data (Section 4.1), then move onto realistic settings. For all experiments, we use $\Pi$-Nets based on the product of polynomials formulation (Appendix), in the same vein as [16].

---

[2]We offer a brief characterization of the result in the Appendix.

### 4.1 EXPERIMENTS ON SYNTHETIC DATA

Our first experiment relates to learning spherical harmonics in the approximately infinite width NTK regime [12]. More precisely, the task comprises of learning linear combinations of spherical harmonics with data uniformly distributed on the unit sphere. We consider large-width two-layer networks to simulate the infinite-width settings and we compare the performance of Π-Nets against standard neural networks. The optimization method is full batch vanilla gradient descent, to avoid stochastic effects. The observations align with our theoretical findings, we defer the results to Appendix due to space constraints. Next, we assess whether the spectral bias manifests beyond the NTK regime. We consider the task of learning sinusoidal signals with smaller width networks and larger learning rates, so as to expressly violate the NTK assumptions.

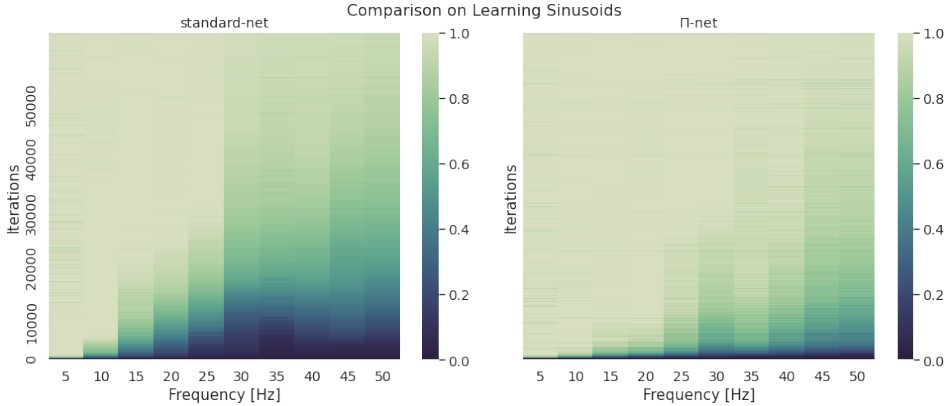

Figure 1: Comparison of learning speeds across different frequencies. The target signal. i.e., sinusoids, is transformed in the Fourier domain and the learned components are compared to the true amplitudes. On the colormap scale, 1 denotes the perfect approximation. We observe that the Π-Net (right) learns higher frequencies faster, i.e., lower in the y axis, than the standard network (left).

**Learning Sinusoids** The goal is to learn sinusoidal signals [37]. Given frequencies $k_i \in \mathcal{K}$, with amplitudes $A_i \in \alpha$ and phases $\phi_i \in \Phi$, the target map $f^* : [0, 1] \to \mathbb{R}$ is defined as $f^*(x) = A_i \sin(2\pi k_i x + \phi_i)$. We approximate this map with two methods: i) a fully connected neural network and ii) a Π-Net. In the first setting, we compare a 256-unit wide, six-layer deep neural network against a 256-unit wide, six-layer deep Π-Net (which has five multiplicative layers). In each case, the network $f_\theta$ (parametrized by $\theta$) regresses over $f^*$ ($\mathcal{K} = (5, 10, ..., 45, 50)$, $\phi_i \sim U(0, 2\pi)$ and $A_i = 1 \ \forall \ i$), using $N = 200$ evenly spaced input samples over $[0, 1]$ and with a fixed learning rate (same for both networks); the spectrum of the network $\tilde{f}_\theta(k)$ is monitored during training by tracking the magnitude of learned components for each frequency, averaged over 5 runs (Fig. 1).

**Remark 6** The experimental setup for training the networks replicates the setup of Rahaman et al. [37], without any special initialization or hyper-parameter tuning for either network and the Π-Net is implemented exactly as the product of polynomials formulation specified in the Appendix.

We observe that Π-Nets do speed up training of higher frequencies. To better substantiate our claim, we consider two additional variants, the results of which are included in the Appendix. In the first, we compare a deeper nine-layer feedforward network with a six-layer Π-Net with only three multiplicative layers, such that the feedforward network includes more parameters (Fig. 10). Since skip connections (additive) are known to speed-up training, we next compare the Π-Net with a feedforward network of same depth, but with additive skip connections instead of multiplicative (Fig. 11). In both cases, we verify that the speed-up in learning higher frequencies due to multiplicative layers is much more significant.

**Discussion**: In the task of learning sinusoids, multiplicative interactions can speed up the learning of higher frequencies and are more effective at doing so than simply increasing the depth. In the Appendix, we conduct an experiment to evaluate the robustness of the networks in retaining high frequency information (Fig. 12) and we see that Π-Nets are more robust to perturbations in the higher frequencies. Remarkably, the Π-Net with more interactions is noticeably more robust than the lower degree polynomial network (Fig. 13 in the appendix). We hypothesize that Π-Nets enhance the

representational space for higher frequencies, relating to our observations on the RKHS in the NTK regime, which makes them less susceptible than standard neural networks. This could also explain why more multiplicative interactions lead to more robustness.

## 4.2 LEARNING IMAGES

To further validate our claim in natural images, we adopt the convolutional layers often used in deep convolutional neural networks (DCNNs) [31]. DCNNs are ubiquitous in vision, partly due to the DCNN structure imposing a suitable prior for tasks with natural images, termed as the Deep Image Prior (DIP) [33]. We precisely assess how this prior changes with multiplicative interactions in Π-Nets in a denoising setup adapted from the DIP framework.

**Experiment - Denoising**: We consider an image $x \in \mathbb{R}^{3 \times H \times W}$ and obtain its noisy version $x_0 = x + \delta$, perturbed with Gaussian noise. For the input, we sample a random tensor $z \in \mathbb{R}^{N \times H \times W}$ (N = 32 in our setup). We consider a neural network $f_\theta$ (making the parametrization by $\theta$ explicit), and optimize the reconstruction loss, $\min_\theta ||f_\theta(z) - x_0||^2$, with respect to the noisy image. We can expect the DCNN structure to first learn the ("natural") features corresponding to the true image $x$ and pick up the noise only in the latter stages, which can then be avoided using early-stopping [33].

**Remark 7** Our goal is not to quantify the denoising performance but rather, to validate the experimental observation on sinusoids and assess whether Π-Nets can speed up learning of high frequencies in real-world applications. If the speed up is confirmed, Π-Nets can be early-stopped, i.e., require less iterations for achieving the target result.

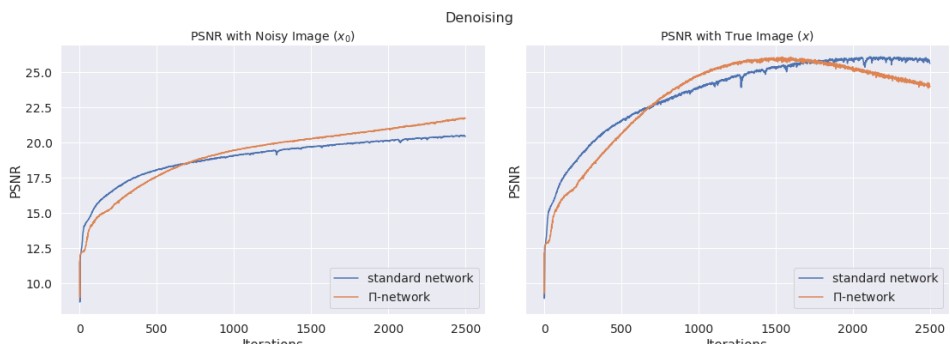

Figure 2: The plots track the progress of the denoising task in the DIP setup via measuring the PSNR w.r.t. the noisy image (left) and the true image (right). We ideally want to stop the optimization process when the PSNR w.r.t the true image is maximum. We observe that for Π-Nets, the maximum PSNR point occurs much earlier, beyond which the PSNR w.r.t the true image starts to decrease as the network begins to learn the noise. This indicates that the Π-Net shows a reduced impedance towards high frequency information, compared to standard networks.

As in [37], we experiment with the U-net architecture, adapting it suitably for Π-Nets with an image of resolution 480 x 600. The implementation details are included in the Appendix. We train both networks for 2500 iterations, with the same learning rate and identical input tensor $z$. We monitor the Peak Signal-to-Noise Ratio (PSNR) curves while training, in Fig. 2. The peak-PSNR with the true image for both standard and Π-Nets are roughly the same, indicating similar denoising performance. However, Π-net achieves this peak PSNR in approximately half the number of iterations. Beyond this point, the Π-net PSNR with the true image decreases as it starts to pick up the high-frequency noise. We confirm this in the visual snapshot of the output of the two networks (Fig. 16 in the appendix). We conclude that Π-Nets do indeed speed up learning in images, by showing a reduced impedance towards high-frequency information. Next, we check how this bias affects learning high frequency information pertaining to natural images in the absence of noise.

**Experiment - Power Spectrum Analysis**: We consider the identical setup as the denoising experiment but without the noise perturbation, setting $\delta = 0$. Therefore, the task for the network $f_\theta$ learn $\theta^*$ such that $f_{\theta*}(z) = x$. We allow 600 iterations of gradient descent and we monitor the radial power spectral density (p.s.d.) of the network output image against the ground truth p.s.d. It allows us to explicitly track the learning progress for each the frequency magnitude.

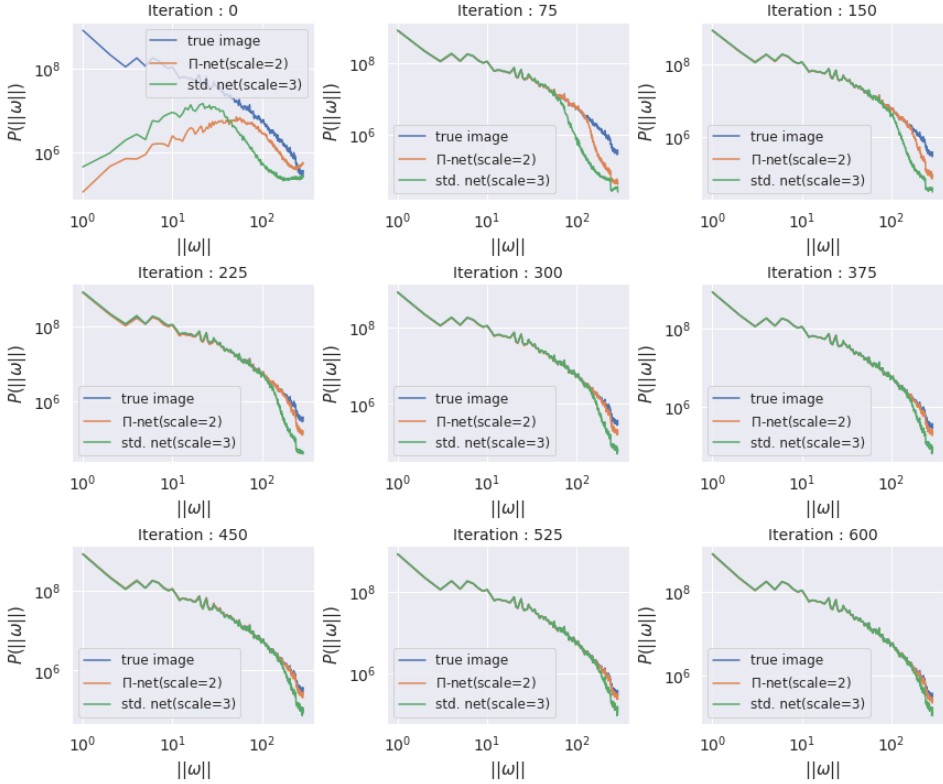

Figure 3: We compare the power spectral density curves at different checkpoints during 600 iterations of optimization, for standard (scale = 3) vs Π networks (scale = 2). The goal for each network is to match the power spectral density of the target image. We observe that the Π-Net picks up high-frequency information in the image faster.

We use the same U-net structure and suitably modify it for Π-Nets as before. Since the multiplicative interactions introduce more parameters, we reduce the depth/scale (by scale, we mean number of down/up-sampling operations in the U-net) of Π-Net to ensure roughly the same parameters (∼1.3 million). The learning speeds in different frequencies can be observed in Fig. 3. In the Appendix, we repeat the same experiment for equal depth networks.

**Discussion**: From the preceding two experiments, we establish how Π-Nets pick up high frequency information faster than (larger) DCNNs. In the Appendix, we note the effect of multiplicative interactions on the deep image prior of the network. We also verify that this altered spectral bias remains relevant for standard learning tasks with natural images. Consequently, for tasks where capturing the finer details of the image is important (such as the denoising instance above), this potentially leads to a reduced number of iterations for Π-Nets as compared to standard networks. We expect the insights from our work to be useful for guiding network design with multiplicative interactions in large-scale experiments.

### 4.3 EFFECT OF LABEL NOISE IN CLASSIFICATION

Finally, we conduct an experiment in a classification setting, wherein the goal is to quantify the effect of this spectral bias in presence of label noise. The setup is adapted from Rahaman et al. [37], a fully connected 6-layer deep 256-unit wide network is trained on a binary classification on MNIST images (by only considering classes "3" and "8"). The labels are perturbed by label noise of different frequencies and the network is trained on the noisy labels with mean squared loss. Rahaman et al. [37] noted for feedforward networks that for a fixed amplitude, low frequency label noise degrades generalization performance (difference between training and validation losses) to a larger extent than high frequency noise. While the low frequency noise is learned instantly, the high-frequency noise is only fit later in the training. As a result, the network learns only true labels at first, and this corresponds to a "dip" in the true validation loss in the early stages. This "dip" becomes larger

with increasing frequency of noise, indicating network impedance towards higher frequencies in the early iterations. Thus, it is only during the latter stages that the true validation loss degrades.

We repeat the experiment with Π-Nets, by supplementing the feedforward network with exactly one multiplicative layer, and contrast the two networks. Since Π-Nets pick up high frequency variations faster, we consequently expect a smaller "dip". We train for 5000 iterations with identical learning rates and observe the validation loss curves for different label noise frequencies. In Fig. 4, we zoom in on the first 1000 iterations for better visualization (complete curves included in the Appendix).

**Discussion**: While the performance for the two networks is identical in absence of label noise (freq=0), the validation "dip" in Π-Net for higher frequencies (0.3, 0.5, 1) is visibly smaller and is negligible for lower frequencies (0.1, 0.2), indicating that Π-Nets pick up the high frequency label noise in the decision boundaries much faster. Additionally, we verify that increasing the number of multiplicative layers reduces the "dip" even further. The plots are deferred to the Appendix.

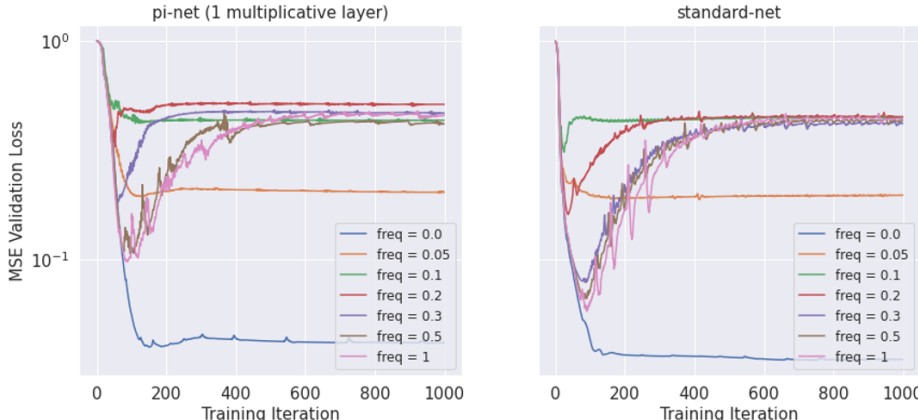

Figure 4: We compare the validation loss curves for the first 1000 iterations on the binary classification task, for Π-Net (left) and standard network (right). For the same frequency, the validation dips for Π-Net are much smaller, indicating a higher tendency to pick up high-frequency label noise.

## 5 DISCUSSION

In this work, we focus on the spectral of polynomial neural networks. Our theoretical results in the NTK regime utilize a two-layer polynomial network and demonstrate a speed-up in the learning of higher frequencies over standard neural networks. We experimentally verify these properties, even outside of the NTK training regime. Additionally, the results offer intuition behind the success of networks that use multiplicative interactions, such as StyleGAN [28], whose connections to polynomials have been noted previously [17].

As a future direction, we aim to design controlled settings to study how polynomial and multiplicative interactions affect performance in state-of-the-art conditional generative models for image generation or deblurring, where high-frequency information is critical for photo-realistic outcomes. It should be noted that the spectral bias of standard neural networks towards low frequency (complexity) functions is believed to help in generalization. Therefore, another important direction is to explore whether this enhanced spectral bias of polynomials towards higher complexity functions translates to differences in their generalization properties. Finally, our results in the classification task yield a further research direction towards analyzing the smoothness of decision boundaries of Π-Nets with multiplicative interactions, especially focusing on areas wherein this effect becomes relevant such as adversarial susceptibility or knowledge distillation.

## ACKNOWLEDGMENTS

We are thankful to the anonymous reviewers for their constructive feedback. Research was sponsored by the Army Research Office and was accomplished under Grant Number W911NF-19-1-0404. This project has received funding from the European Research Council (ERC) under the

European Union's Horizon 2020 research and innovation programme (grant agreement n° 725594 - time-data). JM was supported by the ERC grant number 714381 (SOLARIS project) and by ANR 3IA MIAI@Grenoble Alpes, (ANR19-P3IA-0003).

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

APPENDIX

## A  A PRIMER ON POLYNOMIAL NEURAL NETWORKS ($\Pi$-NETS)

Polynomial neural networks ($\Pi$-Nets) [16] were recently introduced with the output being a high degree polynomial expansion of the input. The parameters of $\Pi$-Nets can be represented as higher order tensors. These networks are able to serve as function approximators even without the use of non-linear activations. When used in conjunction with activations, the models demonstrate better expressivity and achieve state-of-art results on a variety of learning tasks. We next describe the construction of $\Pi$-Nets.

### A.1  METHOD

**Mode-m vector product**: Consider an $M^{th}$ order tensor $\boldsymbol{\mathcal{X}}$, with each of its element addressed by M indices, i.e., $(\boldsymbol{\mathcal{X}})_{i_1,i_2,\ldots,i_M} = x_{i_1,i_2,\ldots,i_M}$ . An $M^{th}$-order real-valued tensor X is defined over the tensor space $\mathbb{R}_1^I \times I_2 \times \ldots \times I_M$ , where $I_m \in \mathbb{Z}$ for $m = 1, 2, \ldots M$. The *mode-m* unfolding of a tensor $\boldsymbol{\mathcal{X}} \in \mathbb{R}^{I_1 \times I_2 \times \ldots \times I_M}$ maps $\boldsymbol{\mathcal{X}}$ to a matrix $\boldsymbol{X}_{(\boldsymbol{m})} \in \mathbb{R}^{I_m \times \hat{I}_m}$ with $\hat{I}_m = \prod_{i=1,i\neq m}^{M} I_i$. such that the tensor element $\boldsymbol{\mathcal{X}}_{i_1,i_2,\ldots,i_M}$ is mapped to the matrix element $\boldsymbol{X}_{i_m,j}$ where $j = 1 + \sum_{k=1,k\neq m}^{M}(i_k - 1)J_k$, where $J_k = \prod_{n=1,n\neq m}^{M} I_n$. The *mode-m* vector product of $\boldsymbol{\mathcal{X}}$ with a vector $\boldsymbol{u} \in \mathbb{R}^{I_m}$, denoted by $X \times_m \boldsymbol{u} \in \mathbb{R}^{I_1 \times I_2 \times \ldots \times I_{m-1} \times I_{m+1} \ldots I_M}$ , results in a tensor of order $M - 1$:

$$(\boldsymbol{\mathcal{X}} \times_m \boldsymbol{u})_{i_1,\ldots,i_{m-1},i_{m+1},\ldots,i_M} = \sum_{i_m=1}^{I_m} x_{i_1,i_2,\ldots,i_M} u_{i_m}.$$

Furthermore, we denote $\boldsymbol{\mathcal{X}} \times_1 \boldsymbol{u}_1 \times_2 \boldsymbol{u}_2 \ldots \times_M \boldsymbol{u}_M = \boldsymbol{\mathcal{X}} \prod_{i=1}^{m} \times_m \boldsymbol{u}_m.$

$\Pi$-Net learns a function $G : \mathbb{R}^d \to \mathbb{R}^o$, such that each element of the output $x_j$ can be expressed as a polynomial of all the input elements $z_i$, with $i \in [1, d]$ as follows:

$$x_j = G(\boldsymbol{z})_j = \beta_j + \boldsymbol{w}_j^{[1]^T} \boldsymbol{z} + \boldsymbol{z}^T \boldsymbol{W}_j^{[2]} \boldsymbol{z} +$$
$$\boldsymbol{\mathcal{W}}_j^{[3]} \times_1 \boldsymbol{z} \times_2 \boldsymbol{z} \times_3 \boldsymbol{z} + \cdots + \boldsymbol{\mathcal{W}}_j^{[N]} \prod_{n=1}^{N} \times_n \boldsymbol{z}, \tag{10}$$

where $\beta_j \in \mathbb{R}$ and $\left\{ \boldsymbol{\mathcal{W}}_j^{[n]} \in \mathbb{R}^{\prod_{m=1}^{n} \times_m d} \right\}_{n=1}^{N}$ are parameters for approximating the output $x_j$. The correlations (of the input elements $z_i$) up to $N^{th}$ order emerge in Eq. (10). More compactly:

$$\boldsymbol{x} = G(\boldsymbol{z}) = \sum_{n=1}^{N} \left( \boldsymbol{\mathcal{W}}^{[n]} \prod_{j=2}^{n+1} \times_j \boldsymbol{z} \right) + \beta, \tag{11}$$

where $\beta \in \mathbb{R}^o$ and $\left\{ \boldsymbol{\mathcal{W}}^{[n]} \in \mathbb{R}^{o \times \prod_{m=1}^{n} \times_m d} \right\}_{n=1}^{N}$ are the learnable parameters. This form allows us to approximate any smooth function as per an extension of the Weierstrass Theorem. To prevent an exponential number of parameters, the authors propose using coupled tensor decompositions.

### A.2  TENSOR DECOMPOSITION FOR SINGLE POLYNOMIAL

An appropriate tensor decomposition on the parameters in Eq. (11) allows for implementation with a neural network. Here, we briefly describe one such decomposition:

**Model: NCP**: Next, we consider a joint hierarchical decomposition on the polynomial parameters. A Nested coupled CP decomposition (NCP) results in the following recursive relationship for $N^{th}$ order approximation:

$$\boldsymbol{x}_n = \left( \boldsymbol{A}_{[n]}^T \boldsymbol{z} \right) * \left( \boldsymbol{S}_{[n]}^T \boldsymbol{x}_{n-1} + \boldsymbol{B}_{[n]}^T \boldsymbol{b}_{[n]} \right), \tag{12}$$

for $n = 2, \ldots, N$ with $\boldsymbol{x}_1 = \left(\boldsymbol{A}_{[n]}^T \boldsymbol{z}\right) * \left(\boldsymbol{B}_{[n]}^T \boldsymbol{b}_{[n]}\right)$ and $\boldsymbol{x} = \boldsymbol{C}\boldsymbol{x}_N + \boldsymbol{\beta}$. The parameters $\boldsymbol{C} \in \mathbb{R}^{o \times k}, \boldsymbol{A}_{[n]} \in \mathbb{R}^{d \times k}, \boldsymbol{S}_{[n]} \in \mathbb{R}^{k \times k}, \boldsymbol{B}_{[n]} \in \mathbb{R}^{\omega \times k}, \boldsymbol{b}_{[n]} \in \mathbb{R}^{\omega}$ for $n = 1, \ldots, N$, are learnable. $\left\{\boldsymbol{b}_{[n]} \in \mathbb{R}^{\omega}\right\}_{n=1}^{N}$ act as a scaling factor for each parameter tensor, whose role is illustrated in case of the third order approximation in Eq. (13):

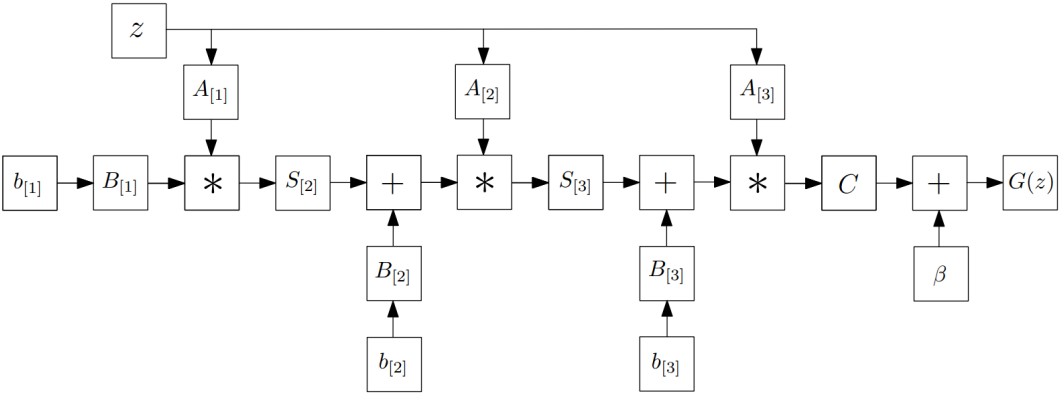

Figure 5: Schematic illustration of the NCP [16].

$$G(\boldsymbol{z}) = \boldsymbol{\beta} + \mathcal{W}^{[1]} \times_2 \boldsymbol{b}_{[1]} \times_3 \boldsymbol{z} + \mathcal{W}^{[2]} \times_2 \boldsymbol{b}_{[2]} \times_3 \boldsymbol{z} \times_4 \boldsymbol{z} + \mathcal{W}^{[3]} \times_2 \boldsymbol{b}_{[3]} \times_3 \boldsymbol{z} \times_4 \boldsymbol{z} \times_5 \boldsymbol{z}. \quad (13)$$

Further, the joint factorization of the parameter tensors (in matrix form) for the third order NCP polynomial may be summarized as follows:

- First order parameters : $\boldsymbol{W}_{(1)}^{[1]} = \boldsymbol{C}(\boldsymbol{A}_{[3]} \odot \boldsymbol{B}_{[3]})^T$.

- Second order parametes: $\boldsymbol{W}_{(1)}^{[2]} = \boldsymbol{C}\left\{\boldsymbol{A}_{[3]} \odot \left[\left(\boldsymbol{A}_{[2]} \odot \boldsymbol{B}_{[2]}\right)\boldsymbol{S}_{[3]}\right]\right\}^T$.

- Third order parameters: $\boldsymbol{W}_{(1)}^{[3]} = \boldsymbol{C}\left\{\boldsymbol{A}_{[3]} \odot \left[\left(\boldsymbol{A}_{[2]} \odot \left\{\left(\boldsymbol{A}_{[1]} \odot \boldsymbol{B}_{[1]}\right)\boldsymbol{S}_{[2]}\right\}\right)\boldsymbol{S}_{[3]}\right]\right\}^T$

### A.3 PRODUCT OF POLYNOMIALS

The second scheme of implementation approximates the target function using a product of polynomials form, wherein the output of first polynomial is fed to the next and so on. The concept is visually depicted in 6; for instance, if each polynomial is degree two, then stacking $N$ such polynomials results in an overall order of $2^N$.

**Remark**: In practice, each matrix operation in the recursive formulation of Π-Nets represents an affine transform on a vector. Therefore, the implementation in standard libraries may be as simple as using a convolutional or a fully-connected layer. The other difference arises due to the multiplicative layers, which may be seen as special 'skip' connections which are combined with the network output via the element-wise vector product (Hadamard product), as opposed to addition. Therefore, it is easy to see why these networks scale as well as standard deep networks.

$$\text{Order } 2^N$$

Figure 6: Schematic illustration of the product of polynomials model [16].

## B   EIGENVALUES DETERMINE THE SPEED OF LEARNING

As noted previously, the integral operator with respect to kernel function is defined as follows:

$$L_\kappa(f)(\boldsymbol{x}) = \int_X \kappa(\boldsymbol{x}, \boldsymbol{y}) f(\boldsymbol{y}) d\tau(\boldsymbol{y}). \tag{14}$$

The key idea in [12] is to establish guarantees on the speed of convergence of gradient descent along different directions of $L_\kappa$, as defined by its eigenfunctions. Consider $\{\lambda_i\}_{i\geqslant 1}$ with $\lambda_1 \geqslant \lambda_2 \geqslant ..$ be the strictly positive eigenvalues of $L_\kappa$ with $\{\phi_i\}_{i\geqslant 1}$ the respective eigenfunctions and define $v_i = n^{-1}(\phi(\boldsymbol{x_1}), \phi(\boldsymbol{x_2}).......\phi(\boldsymbol{x_n}))$. Since the eigenvalues may not be distinct, define $r_k$ as the sum of multiplicities of the first $k$ distinct eigenvalues of $L_\kappa$. Define $V_{r_k} = (v_1..., v_{r_k})$. By definition, $v_{ii\leqslant r_k}$ are rescaled restrictions of orthonormal functions on the training examples. Therefore, they form a set of almost orthonormal bases in the vector space $\mathbb{R}^n$.

Finally, denote $\boldsymbol{y} = (y_1, y_2.....y_n)^T$ and $\hat{\boldsymbol{y}}^{(t)} = (f_{\boldsymbol{W}^{(t)}}(\boldsymbol{x_1}), .....f_{\boldsymbol{W}^{(t)}}(\boldsymbol{x_n}))^T$ as the ground truth and predictions at time $t$, respectively, where $f_{\boldsymbol{W}^{(t)}}$ denotes the 2-layer ReLU network. Then the following result holds:

**Theorem 4.2** [12]: Suppose $|\phi_j(\boldsymbol{x})| \leqslant M$ for $j \in [r_k]$ and $\boldsymbol{x} \in \mathbb{S}^{d+1}$. For any $\epsilon, \delta \geqslant 0$ and integer $k$, if $n \geqslant \Omega(\epsilon^{-2} \cdot \max\{(\lambda_{r_k} - \lambda_{r_k+1})^{-2}, M^4 r_k^2\}), m \geqslant \Omega(poly(T, \tilde{\lambda}_{r_k}^{-1}, \epsilon^{-1}))$, then with probability at least $1 - \delta$, gradient descent with step size $\eta = \tilde{\mathcal{O}}(m^{-1})$ satisfies:

$$n^{-1/2} \cdot ||V_{r_k}^T(\boldsymbol{y} - \hat{\boldsymbol{y}}^{(t)})||_2 \leqslant 2 \cdot (1 - \lambda_{r_k})^T . n^{-1/2} \cdot ||V_{r_k}^T \boldsymbol{y}||_2 + \epsilon, \tag{15}$$

which is to say that the convergence rate of $||V_{r_k}^T(\boldsymbol{y} - \hat{\boldsymbol{y}}^{(t)})||_2$, or alternatively the projection of the residual error on the space spanned by the first $r_k$ eigenvalues is controlled by the $r_k^{\text{th}}$ eigenvalue $\lambda_{r_k}$. With some additional work, this result can be extended to the 2-layer $\Pi$-Netby accounting for the extra width factor of $\Omega(\sqrt{\log m})$, as noted previously.
The key takeaway in that with a wide enough network and large enough sample size, gradient descent first learns the target function along the eigen-directions with larger eigenvalues. Since the decay of eigenvalues is slower for the 2-layer $\Pi$-Net, it leads to a speed-up in learning higher frequencies.

## C   PROOF OF THEOREM 1: DERIVING THE $\Pi$-KERNEL

In this section, we prove our first main result for the $\Pi$-kernel corresponding to theorem 1. Consider a two-layer $\Pi$-Net $f_{\boldsymbol{W}}$ parametrized as follows :

$$f_{\boldsymbol{W}}(\boldsymbol{x}) = \sqrt{\frac{2}{m}} \boldsymbol{W_3}[\sigma(\boldsymbol{W_2}\boldsymbol{x}) * \sigma(\boldsymbol{W_1}\boldsymbol{x})], \tag{16}$$

where the weights $\boldsymbol{W_1} \in \mathbb{R}^{m\times d}, \boldsymbol{W_2} \in \mathbb{R}^{m\times d}, \boldsymbol{W_3} \in \mathbb{R}^{1\times m}$ are initialized with independent identically distributed $\mathcal{N}(0, 1)$ coordinates.

Recall that the neural tangent kernel $\kappa_\pi$ corresponds to the limit of the following inner product

$$\kappa_\pi(\boldsymbol{x}, \boldsymbol{x}') = \lim_{m\to\infty} \langle \nabla_{\boldsymbol{W}} f_{\boldsymbol{W}^{(0)}}(\boldsymbol{x}), \nabla_{\boldsymbol{W}} f_{\boldsymbol{W}^{(0)}}(\boldsymbol{x}') \rangle. \tag{17}$$

We can compute the inner-product Eq. (17) by computing the derivatives with respect to $\boldsymbol{W_1}, \boldsymbol{W_2}$, $\boldsymbol{W_3}$ of $f_{\boldsymbol{W}}$ separately and sum up the inner products to obtain $\kappa_\pi$, since the gradient can be split into three blocks.

We denote by $\tilde{\boldsymbol{\alpha}} := \boldsymbol{W}_1 \boldsymbol{x}$, and by $\tilde{\boldsymbol{\beta}} := \boldsymbol{W}_2 \boldsymbol{x}$ the pre-activation vectors; and by $\boldsymbol{\alpha}, \boldsymbol{\beta}$ the post-activation vectors where the element-wise activation $\sigma$ is applied to $\tilde{\boldsymbol{\alpha}}$ and $\tilde{\boldsymbol{\beta}}$ respectively.

First consider the derivative w.r.t $\boldsymbol{W_1}$ denoted $\partial_{\mathbf{W}_1}$:

$$\partial_{W_1} \boldsymbol{f_W}(\boldsymbol{x}) = \sqrt{\frac{2}{m}} \sum_{i=1}^{m} \mathbf{W}_3^i \sigma(\tilde{\beta}_i(\boldsymbol{x})) \sigma'(\tilde{\alpha}_i(\boldsymbol{x})) \partial_{W_1} \tilde{\alpha}_i(\boldsymbol{x}). \tag{18}$$

Since $\tilde{\alpha}_i = \boldsymbol{e}_i^T \mathbf{W}_1 \boldsymbol{x} = \langle \mathbf{W}_1, \boldsymbol{e}_i \boldsymbol{x}^T \rangle$, where $\boldsymbol{e}_i \in \mathbb{R}^m$ is the i-th canonical basis vector of $\mathbb{R}^m$, we have that

$$\partial_{W_1} \tilde{\alpha}_i(\boldsymbol{x}) = \boldsymbol{e}_i \boldsymbol{x}^T.$$

The contribution to the NTK (Eq. (17)) corresponds to $\langle \partial_{W_1} f_{\mathbf{W}}(\boldsymbol{x}), \partial_{\mathbf{W}_1} \boldsymbol{f_W}(\boldsymbol{x}') \rangle$. Note that since the network is symmetric in $\{\mathbf{W}_1, \mathbf{W}_2\}$, we need only look at $\mathbf{W}_1$ to obtain the contribution for $\mathbf{W}_2$ as well. We find that

$$
\begin{aligned}
\langle \partial_{W_1} f_{\mathbf{W}}(\boldsymbol{x}), \partial_{\mathbf{W}_1} f_{\mathbf{W}}(\boldsymbol{x}') \rangle &= \frac{2}{m} \sum_{i,j=1}^{m} \mathbf{W}_3^i \mathbf{W}_3^j [\sigma(\tilde{\beta}_i(\boldsymbol{x})) \sigma(\tilde{\beta}_j(\boldsymbol{x}'))][\sigma'(\tilde{\alpha}_i(\boldsymbol{x})) \sigma'(\tilde{\alpha}_j(\boldsymbol{x}'))] \langle \boldsymbol{e}_i \boldsymbol{x}^T, \boldsymbol{e}_j \boldsymbol{x}'^T \rangle \\
&= \frac{2}{m} \sum_{i,j=1}^{m} \mathbf{W}_3^i \mathbf{W}_3^j [\sigma(\tilde{\beta}_i(\boldsymbol{x})) \sigma(\tilde{\beta}_j(\boldsymbol{x}'))][\sigma'(\tilde{\alpha}_i(\boldsymbol{x})) \sigma'(\tilde{\alpha}_j(\boldsymbol{x}'))] \boldsymbol{x}^T \boldsymbol{x}' \, \delta_{ij} \\
&= \frac{2}{m} \sum_{i=1}^{m} \mathbf{W}_3^i \mathbf{W}_3^i [\sigma(\tilde{\beta}_i(\boldsymbol{x})) \sigma(\tilde{\beta}_i(\boldsymbol{x}'))][\sigma'(\tilde{\alpha}_i(\boldsymbol{x})) \sigma'(\tilde{\alpha}_i(\boldsymbol{x}'))][\boldsymbol{x}^T \boldsymbol{x}'].
\end{aligned}
\tag{19}
$$

As $\lim m \to \infty$, the above quantity converges to its expectation by the law of large numbers because the terms are independent and identically distributed with finite expectation. Consequently,

$$\lim_{m\to\infty} \langle \partial_{W_1} \boldsymbol{f_W}(\boldsymbol{x}), \partial_{\mathbf{W}_1} \boldsymbol{f_W}(\boldsymbol{x}') \rangle = 2 \langle \boldsymbol{x}, \boldsymbol{x}' \rangle \kappa_1(\boldsymbol{x}, \boldsymbol{x}') \kappa_2(\boldsymbol{x}, \boldsymbol{x}') \tag{20}$$

since

$$
\begin{cases}
\boldsymbol{E}_{\mathbf{W}_1}[\sigma'(\tilde{\alpha}_i(\boldsymbol{x})) \sigma'(\tilde{\alpha}_i(\boldsymbol{x}'))] &= \kappa_1(\boldsymbol{x}, \boldsymbol{x}') \\
\boldsymbol{E}_{\mathbf{W}_2}[\sigma(\tilde{\beta}_i(\boldsymbol{x})) \sigma(\tilde{\beta}_i(\boldsymbol{x}'))] &= \kappa_2(\boldsymbol{x}, \boldsymbol{x}') \\
\boldsymbol{E}_{\mathbf{W}_3}[\mathbf{W}_3^i \mathbf{W}_3^i] &= 1.
\end{cases}
$$

By symmetry, we obtain with a similar term for $\boldsymbol{W}_2$ and their total contribution to Eq. (17) adds up to twice the term obtained for $\boldsymbol{W}_1$. Characterizing the kernel w.r.t $\boldsymbol{W}_3$ is more straightforward:

$$\partial_{\mathbf{W}_3} f_{\boldsymbol{W}}(\boldsymbol{x}) = \sqrt{\frac{2}{m}} \sigma(\tilde{\alpha}(\boldsymbol{x})) * \sigma(\tilde{\beta}(\boldsymbol{x})). \tag{21}$$

Therefore, its contribution to the NTK is

$$\langle \partial_{\mathbf{W}_3} f_{\boldsymbol{W}}(\boldsymbol{x}), \partial_{\mathbf{W}_3} f_{\boldsymbol{W}}(\boldsymbol{x}') \rangle = \frac{2}{m} \sum_{i}^{m} \sigma(\tilde{\alpha}_i(\boldsymbol{x})) \sigma(\tilde{\beta}_i(\boldsymbol{x})) \sigma(\tilde{\alpha}_i(\boldsymbol{x}')) \sigma(\tilde{\beta}_i(\boldsymbol{x}')). \tag{22}$$

Applying the law of large numbers again, as $\lim m \to \infty$, this quantity tends to:

$$E_{\mathbf{W}_1, \mathbf{W}_2 \sim N(\mathbf{0}, \boldsymbol{I})} \{ [\sigma(\langle \mathbf{W}_1, \boldsymbol{x} \rangle)) \sigma(\langle \mathbf{W}_1, \boldsymbol{x}' \rangle)] [\sigma(\langle \mathbf{W}_2, \boldsymbol{x} \rangle)) \sigma(\langle \mathbf{W}_2, \boldsymbol{x}' \rangle)] \}$$
$$= E_{\mathbf{W}_1} [\sigma(\langle \mathbf{W}_1, \boldsymbol{x} \rangle)) \sigma(\langle \mathbf{W}_1, \boldsymbol{x}' \rangle)] E_{\mathbf{W}_2} [\sigma(\langle \mathbf{W}_2, \boldsymbol{x} \rangle)) \sigma(\langle \mathbf{W}_2, \boldsymbol{x}' \rangle)] \qquad (23)$$
$$= 2\boldsymbol{\kappa_2}(\boldsymbol{x}, \boldsymbol{x}') \boldsymbol{\kappa_2}(\boldsymbol{x}, \boldsymbol{x}').$$

The theorem follows by summing up $2 \times$ Eq. (20) and Eq. (23).

**Width Requirements**

As noted previously, two-layer Π-Netneeds an the extra factor $\Omega(\sqrt{\log m})$ in terms of width, to stay close to initialization The proof is largely derived from Lemma A.6 in [27], based on ideas first noted in [1] and therefore, we just provide a rough sketch here. For simplicity, consider the set of weights $(\boldsymbol{W_2}, \boldsymbol{W_3})$ fixed at initialization, i.e. at $\boldsymbol{W_2^{(0)}}, \boldsymbol{W_3^{(0)}}$ and note the first-order Taylor expansion for the Π-Net w.r.t $\boldsymbol{W_1}$ as follows:

$$f_{\boldsymbol{W}}(\boldsymbol{x}) \approx f_{\boldsymbol{W}}^0(\boldsymbol{x}) = f_{\boldsymbol{W^{(0)}}}(\boldsymbol{x}) + \langle \nabla_{\boldsymbol{W_1^{(0)}}} f_{\boldsymbol{W^{(0)}}}(\boldsymbol{x}), \boldsymbol{W_1} - \boldsymbol{W_1^{(0)}} \rangle$$

Here $f_{\boldsymbol{W}}^0$ denotes the first order Taylor expansion of f at $\mathbf{W}_0$. We can then expand the RHS as:

$$\sqrt{\frac{2}{m}} \sum_j \mathbf{W}_{3,j}^{(0)} (\sigma(x^T \mathbf{W}_{2,j}^{(0)}) \sigma(x^T \mathbf{W}_{1,j}^{(0)}) + \sigma(x^T \mathbf{W}_{2,j}^{(0)}) \sigma'(x^T \mathbf{W}_{1,j}^{(0)}) x^T (\mathbf{W}_{1,j} - \mathbf{W}_{1,j}^{(0)})).$$

Consequently, we can bound the approximation error as:

$$|f_{\boldsymbol{W}}(\boldsymbol{x}) - f_{\boldsymbol{W}}^0(\boldsymbol{x})| \leqslant \sqrt{\frac{2}{m}} \sum_j |\sigma(x^T \mathbf{W}_{2,j}^{(0)})| |\mathbf{W}_{3,j}^{(0)}| (\sigma(x^T \mathbf{W}_{1,j}) - (\sigma(x^T \mathbf{W}_{1,j}^{(0)}) + \sigma'(x^T \mathbf{W}_{1,j}^{(0)}) x^T (\mathbf{W}_{1,j} - \mathbf{W}_{1,j}^{(0)}))|.$$

Notice that aside from the $|\sigma(x^T \mathbf{W}_{2,j}^{(0)})|$, the error term is exactly the same as that for a standard network. Also note that $P(x^T \mathbf{W}_{2,j} \geqslant \tau) = P(\sigma(x^T \mathbf{W}_{2,j}) > \tau) \leqslant e^{-\tau^2/2}$, for $\tau > 0$ by the sub-gaussian tail bound. To ensure $|\sigma(x^T \mathbf{W}_{2,j}^{(0)})| \leqslant \tau \; \forall \; j$ with probability $1 - \delta$, we use the union bound to obtain the condition on $\tau$ as $m.e^{-\tau^2/2} < \delta$. Consequently, we have that $\tau \sim \Omega(\sqrt{\log m})$, which essentially becomes an additional constant factor in the error term, over the usual error terms from the standard feed-forward network.

# D    PROOF OF THEOREM 2: CHARACTERIZING THE Π-KERNEL EIGENDECAY

Here, we prove the Π-kernel eigenvalue decay rate stated in theorem 2. We first recall some connections between spherical harmonics and Gegenbauer polynomials.

**Definition 1** *For a given $\alpha \in \mathbb{R}$, Gegenbauer (or ultraspherical) polynomials denoted $C_k^\alpha$ : $[-1, 1] \to \mathbb{R}$ are a family of orthogonal polynomials with respect to the weight function $x \mapsto (1 - x^2)^{\alpha - \frac{1}{2}}$, i.e,*

$$\int_{-1}^{1} C_k^\alpha(x) C_\ell^\alpha(x) (1 - x^2)^{\alpha - \frac{1}{2}} = 0,$$

*for $k \neq \ell$.*

**Remark:** Gegenbauer polynomials are a generalization of *Legendre polynomials* which can be recovered by taking $\alpha = \frac{1}{2}$.

The following addition formula expresses Gegenbauer polynomials in terms of spherical harmonics.

**Lemma 1 (Theorem 4.11 in [20])** *(Addition formula) Let $\{Y_{k,j}\}_{j=1}^{N(d,k)}$ denote spherical harmonics of degree $k$ in $d + 1$ variables. It holds that, for any $\boldsymbol{x}, \boldsymbol{x}' \in \mathbb{S}^d$,*

$$\sum_{j=1}^{N(d,k)} Y_{k,j}(\boldsymbol{x}) Y_{k,j}(\boldsymbol{x}') = N(d,k) C_k^{(\frac{d-1}{2})}(\langle \boldsymbol{x}, \boldsymbol{x}' \rangle).$$

By making use of this Lemma, we can rewrite the Mercer decomposition of any admissible dot product kernel $K$ when the data is uniform on the unit sphere. Indeed, by simplifying the Mercer's decomposition in terms of spherical harmonics we have :

$$
\begin{aligned}
K(\boldsymbol{x}, \boldsymbol{x}') &= \sum_{k=0}^{\infty} \mu_k \sum_{j=1}^{N(d,k)} Y_{k,j}(\boldsymbol{x}) Y_{k,j}(\boldsymbol{x}') \\
&= \sum_{k=0}^{\infty} \mu_k N(d,k) C_k^{(\frac{d-1}{2})}(\langle \boldsymbol{x}, \boldsymbol{x}' \rangle),
\end{aligned}
\tag{24}
$$

where $(\mu_k)_{k=0}^{\infty}$ are its corresponding eigenvalues, and the second equality follows from Lemma 1.

Now, since $\boldsymbol{\kappa_1}$ and $\boldsymbol{\kappa_2}$ are dot-product Mercer kernels, we can, akin to Eq. (24), obtain their respective decompositions in terms of Gegenbauer polynomials. There exist $(\mu_{1,k})_{k=0}^{\infty}$ and $(\mu_{2,k})_{k=0}^{\infty}$, both sequences of eigenvalues, such that

$$
\begin{aligned}
\langle \boldsymbol{x}, \boldsymbol{x}' \rangle \boldsymbol{\kappa_1}(\boldsymbol{x}, \boldsymbol{x}') &= \sum_{k=0}^{\infty} \mu_{1,k} N(d,k) C_k^{(\frac{d-1}{2})}(\langle \boldsymbol{x}, \boldsymbol{x}' \rangle) \\
\boldsymbol{\kappa_2}(\boldsymbol{x}, \boldsymbol{x}') &= \sum_{k=0}^{\infty} \mu_{2,k} N(d,k) C_k^{(\frac{d-1}{2})}(\langle \boldsymbol{x}, \boldsymbol{x}' \rangle).
\end{aligned}
\tag{25}
$$

The decay rate of the eigenvalues $(\mu_{1,k})_{k=0}^{\infty}$ and $(\mu_{2,k})_{k=0}^{\infty}$ is known [5, 10, 12] and is stated in the following Lemma.

**Lemma 2 (Appendix D.2 of [5], Lemma 17 of [10])** *For $k \gg d$, $k$ even, we have that $\mu_{1,k} \sim A(d) k^{-d-1}$ and $\mu_{2,k} \sim B(d) k^{-d-2-1/2}$, where $A(d)$ and $B(d)$ are constants only depending on the dimension $d$.*

Our goal is to establish the decay rate of the eigenvalues $(\mu_{\pi,k})_{k=0}^{\infty}$ of the kernel

$$\boldsymbol{\kappa_\pi}(\boldsymbol{x}, \boldsymbol{x}') = 2(2\langle \boldsymbol{x}, \boldsymbol{x}' \rangle \boldsymbol{\kappa_1}(\boldsymbol{x}, \boldsymbol{x}') + \boldsymbol{\kappa_2}(\boldsymbol{x}, \boldsymbol{x}')) \boldsymbol{\kappa_2}(\boldsymbol{x}, \boldsymbol{x}'). \tag{26}$$

By plugging in the representations in Eq. (25) into the Π-kernel expression (Eq. (26)), we find that

$$\boldsymbol{\kappa_\pi}(\boldsymbol{x}, \boldsymbol{x}') = 2\left(\sum_{k=0}^\infty (2\mu_{1,k} + \mu_{2,k}) N(d,k) C_k^{(\frac{d-1}{2})}(\langle \boldsymbol{x}, \boldsymbol{x}'\rangle)\right)\left(\sum_{k=0}^\infty \mu_{2,k} N(d,k) C_k^{(\frac{d-1}{2})}(\langle \boldsymbol{x}, \boldsymbol{x}'\rangle)\right).$$

(27)

Moreover, since we can also write

$$\boldsymbol{\kappa_\pi}(\boldsymbol{x}, \boldsymbol{x}') = \sum_{k=0}^\infty \mu_{\pi,k} N(d,k) C_k^{(\frac{d-1}{2})}(\langle \boldsymbol{x}, \boldsymbol{x}'\rangle),$$

an appropriate factorisation of the product Eq. (27) will allow us to deduce the order of $\mu_{\pi,k}$ by equating the coefficients appearing in front of the Gegenbauer polynomials.

Developing the product in Eq. (27) leads to the appearance of products of Gegenbauer polynomials. The product of two Gegenbauer polynomials turns out to be a linear combination of other Gegenbauer polynomials.

**Lemma 3 (Equation 8 in [13])** *Let $\alpha \in \mathbb{R}$, for any $m, n \in \mathbb{N}$, there exists positive coefficients $(\lambda_s^{(m,n)})_{s=0}^{\min(m,n)}$ such that*

$$C_m^{(\alpha)}(x) C_n^{(\alpha)}(x) = \sum_{s=0}^{\min(m,n)} \lambda_s^{(m,n)} C_{m+n-2s}^{(\alpha)}(x).$$

(28)

By using this expansion, coefficients for each Gegenbauer polynomial may be identified.

Take $k \in \mathbb{N}$ to be an even number. We know from plugging Lemma 3 into Eq. (27) that the coefficient $\mu_{\pi,2k} N(d, 2k)$ in front of the polynomial $C_{2k}^{(\frac{d-1}{2})}$ can be lower bounded as follows.

$$\mu_{\pi,2k} N(d, 2k) \geqslant N(d,k)^2 (2\mu_{1,k} + \mu_{2,k}) \mu_{2,k} \lambda_0^{(k,k)}.$$

(29)

We obtain this lower bound by considering the contribution of a single term in Eq. (28) where $m = n = k$ and $s = 0$ to the coefficient in front of $C_{2k}^{(\frac{d-1}{2})}$. This contribution is a lower bound because all coefficients involved in the product are non-negative.

Consequently, in order to establish a decay rate for $(\mu_{\pi,k})$ it suffices to study the decay rate of $\lambda_0^{(k,k)}$. This rate is given by the following Lemma.

**Lemma 4** *Let $\alpha = \frac{d-1}{2}$ be an integer, the coefficient $\lambda_0^{(k,k)}$ defined in Lemma 3 admits the following expression $\lambda_0^{(k,k)} = \frac{((\alpha+k-1)!)^2}{(\alpha-1)!((k)!)^2} \frac{(2k)!}{(\alpha+2k-1)!}$. Moreover, for $k \gg d$, it holds that*

$$\lambda_0^{(k,k)} \sim k^{(d/2)}.$$

See Appendix D.1 for a proof.

We now dispose of all the necessary results to prove Theorem 2. Starting from Eq. (29), we find that for $k \gg d$, we have that

$$\begin{aligned} \mu_{\pi,2k} &\geqslant \frac{N(d,k)^2}{N(d,2k)} (2\mu_{1,k} + \mu_{2,k}) \mu_{2,k} \lambda_0^{(k,k)} \\ &\sim \frac{k^d k^d}{(2k)^d} (2\mu_{1,k} + \mu_{2,k}) \mu_{2,k} \lambda_0^{(k,k)} \quad \text{(by Stirling)} \\ &= \frac{k^d}{2^d} \Omega(k^{-2d-2}) k^{(d/2)} \quad \text{(by Lemma 2 and Lemma 4)} \\ &= \Omega((2k)^{-d/2-2}). \end{aligned}$$

(30)

This concludes the proof that, for $k$ divisible by 4, we have $\mu_{\pi,k} = \Omega(k^{-d/2-2})$. For $k = 1 \bmod 4$, we conduct the same reasoning taking the coefficient $\lambda_0^{(2k+1,2k)}$. For $k = 3 \bmod 4$, the coefficient to consider is $\lambda_0^{(2k+1,2k+2)}$. The equivalence derivation Lemma 4 proceeds in exactly the same manner for these coefficients.

### D.1 PROOF OF LEMMA 4

Let us denote $(\alpha)_k := \alpha(\alpha + 1)(\alpha + 2)....(\alpha + k - 1)$ and $(\alpha)_0 := 1$.

From Equation 8 in [13], we have that

$$
\begin{aligned}
\lambda_0^{(k,k)} &= \frac{2k + \alpha}{2k + \alpha} \cdot \frac{(\alpha)_0 (\alpha)_k (\alpha)_k}{0!(k)!(k)!} \cdot \frac{(2\alpha)_{2k}}{(\alpha)_{2k}} \cdot \frac{(2k)!}{(2\alpha)_{2k}} \\
&= \frac{(\alpha)_k (\alpha)_k}{(k)!(k)!} \frac{(2k)!}{(\alpha)_{2k}} \\
&= \frac{((\alpha + k - 1)!)^2}{((\alpha - 1)!(k)!)^2} \frac{(2k)!(\alpha - 1)!}{(\alpha + 2k - 1)!} \\
&= \frac{((\alpha + k - 1)!)^2}{(\alpha - 1)!((k)!)^2} \frac{(2k)!}{(\alpha + 2k - 1)!}.
\end{aligned}
\tag{31}
$$

We can apply Stirling's approximation stating that $n! \sim \sqrt{2\pi n}(\frac{n}{e})^n$ to find that:

$$
\lambda_0^{(k,k)} \sim \frac{(\alpha + k - 1)(\alpha + k - 1)^{2(\alpha + k - 1)}}{\sqrt{\alpha - 1}(\alpha - 1)^{\alpha - 1} k(k)^{2k}} \cdot \frac{\sqrt{2k}(2k)^{2k}}{\sqrt{\alpha + 2k - 1}(\alpha + 2k - 1)^{\alpha + 2k - 1}}
\tag{32}
$$

Considering the case when $k \gg \alpha$ or equivalently, $k \gg d$ since $\alpha = (d - 1)/2$, we obtain the following simplification:

$$
\begin{aligned}
\lambda_0^{(k,k)} &\sim \frac{k^{(2v + 2k - 1)}}{k^{2k+1}} \cdot \frac{(2k)^{2k+0.5}}{(2k)^{v + 2k - 0.5}} \\
&\sim (\frac{k}{2})^{\alpha - 1} \sim k^{(d/2)}.
\end{aligned}
\tag{33}
$$

# E LEARNING SPHERICAL HARMONICS WITH Π-KERNEL

Following the setup in [12], we perform a similar experiment on learning combinations of spherical harmonics in the NTK regime. We define and initialize the Π-Net exactly as specified, with a width of 32768 neurons and using vanilla gradient descent, to approximate the kernel learning in the infinite width limit. We take $n = 1000$ samples, $\{\boldsymbol{x}\}_{i=1}^n$ from the uniform distribution on the unit sphere $\mathbb{S}^{10}$. We define our target function with integral $k \in \mathcal{K}$ as follows:

$$f^*(\boldsymbol{x}) = \frac{1}{N(\mathcal{K})} \sum_{k \in K} A_k P_k(\langle \boldsymbol{x}, \zeta_k \rangle), \tag{34}$$

where the $P_k(t)$ is the Gegenbauer polynomial with degree $k$, $\zeta_k$ are fixed vectors that are independently generated from uniform distribution on unit sphere in $\mathbb{R}^{10}$, and $N(\mathcal{K})$ is a suitable normalizing constant to keep the order of magnitude of the target function approximately the same for different choices of $\mathcal{K}$. As noted previously, $f^*$ may be seen as a linear combination of spherical harmonics and we compare the error residuals during the learning process in standard vs Π-Nets in the NTK regime, for varying $\mathcal{K}$. We use a moving average of range 20 on these curves to smoothen out the heavy oscillations in the latter stages.

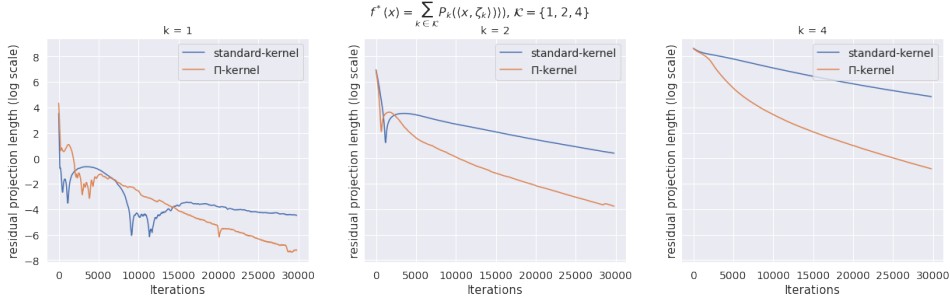

Figure 7

Figure 8: The plots represent a comparison of log-scale convergence curve of error projection lengths for standard vs Π-kernel for different order harmonics with $\mathcal{K} = \{1, 2, 4\}$, indicating a clear improvement in the rate of convergence of error for higher harmonics

For the first setting, we look at $\mathcal{K} = \{1, 2, 4\}$, with corresponding weight ratio as $A_1 : A_2 : A_4 = 1 : 1 : 1$. For each frequency, we look at the rate of convergence for the two kernels.
We observe (Fig. 8) that the rate of error convergence for the Π-kernel is much faster than the standard two-layer NTK and especially for higher harmonics (clearest increase for the highest $k = 4$), which is exactly what is expected from theoretical results.

Next, we consider $\mathcal{K} = \{1, 3, 4, 5, 8, 12\}$ i.e. we consider higher frequency harmonics and also introduce odd harmonics, with the respective $A_k$ assigned the equal weights relative to each other. Recall that the eigenvalues corresponding to odd harmonics vanish for the standard kernel, meaning that we expect a much slower convergence rate for them. As before, we look at the rate of convergence for individual harmonics for the two kernels. The convergence curves are presented in Fig. 9. We verify the speed-up in higher frequencies, and an especially notable gap for odd harmonics other than $k = 1$.

**Discussion**: The empirical results strongly support our hypothesis that the Π-kernel can speed up learning higher harmonics faster than the standard two-layer kernel, even for settings outside of $k \gg d$.

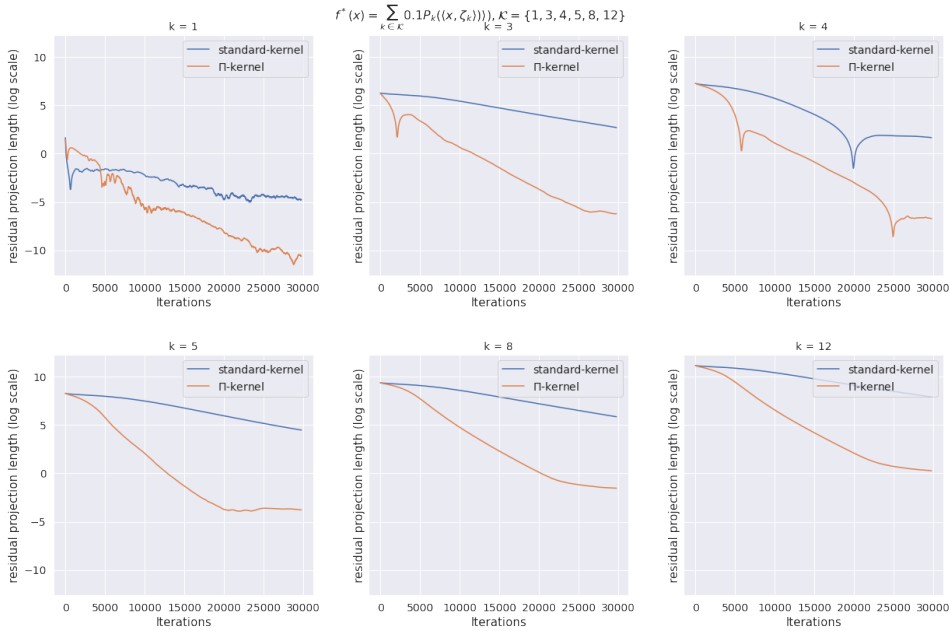

Figure 9: The plots represent a comparison of log-scale convergence curve of error projection lengths for standard vs $\Pi$-kernel for different order harmonics with $\mathcal{K} = \{1, 3, 4, 5, 8, 12\}$. We again see a clear improvement in the rate of convergence of error for higher harmonics, and especially so for odd harmonics greater than 1.

# F    SYNTHETIC EXPERIMENTS WITH SINUSOIDS

## F.1    LEARNING SINUSOIDS

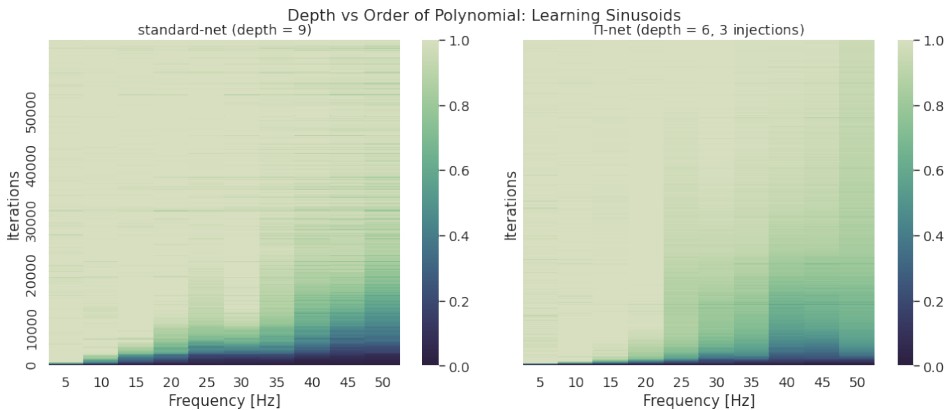

Figure 10: The heat map denotes a comparison on the effectiveness of increasing depth vs introducing multiplicative interactions via Π-Nets for learning high-frequency information. The empirical evidence shows that multiplicative layers are more effective for learning higher frequencies faster.

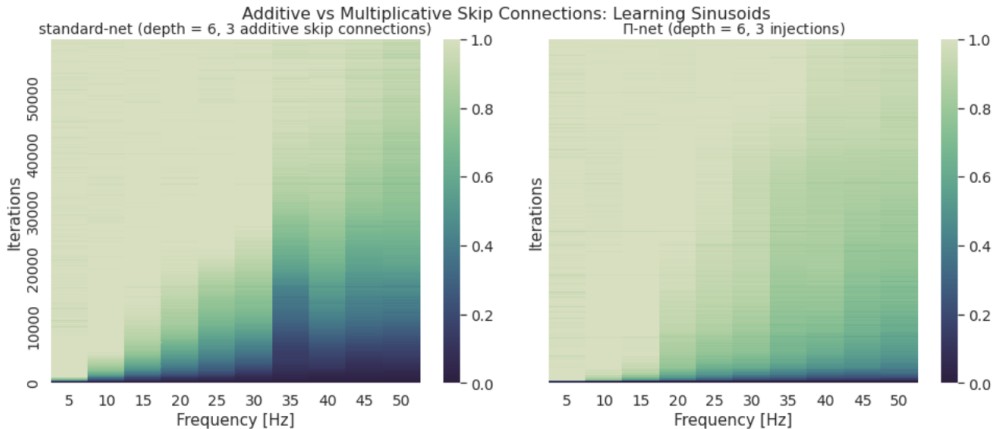

Figure 11: The heat map denotes a comparison on the effectiveness of multiplicative layers via Π-Nets vs only using additive skip connections. The additive skip connections do not seem to affect the spectral bias over standard neural networks in terms of improving speed of learning for high frequencies.

## F.2    ROBUSTNESS

**Robustness to Perturbations**: Motivated by the observations of Rahaman et al. [37], we evaluate the susceptibility of higher frequencies to random perturbations for standard networks and Π-Nets. The setup relies on the learning task in the previous experiment. More precisely, following convergence of the network to a low-error approximation of target $f^*$ (denoted by $f_{\theta*}$), random isotropic perturbations are introduced to the network parameters: $\theta = \theta^* + \delta\hat{\theta}$. We monitor the effect of increasing the perturbation magnitude $\delta$ on the frequencies of interest. In the first setting (Fig. 12), we compare the effects of perturbations on the six-layer standard network to the six-layer Π-Net (with five multiplicative layers). In the second setting (Fig. 13), we directly compare the two variants of 6-layer deep Π-Nets i.e. one with five layers and the other with three layers. We observe that that Π-Nets are more robust to perturbations, especially in terms of retaining high frequency information, as compared to standard feedforward networks. We also note that the presence of more multiplicative interactions makes the network more robust.

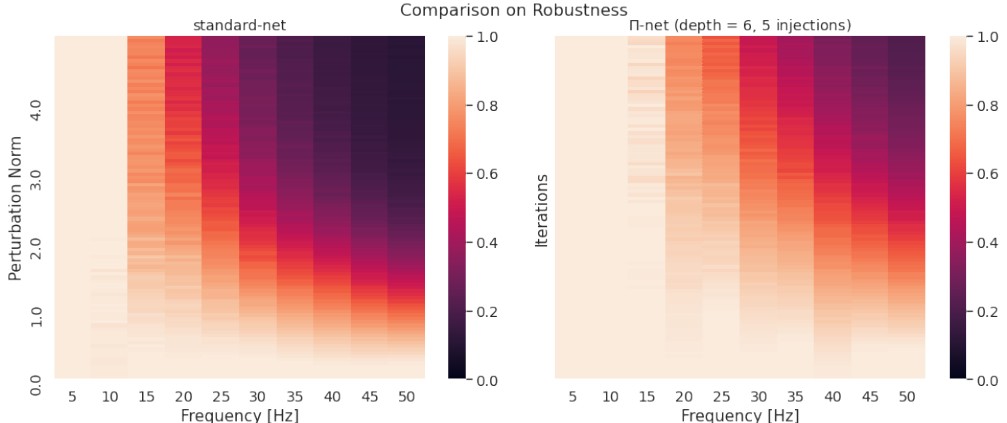

Figure 12: The heap maps present a comparison on the robustness to random parameter perturbations for six-layer standard vs six-layer Π-Net. The y-axis denotes the norm of the random perturbation. For standard networks, the high-frequency information is lost quickly as the perturbation norm increases, while Π-Nets are much more effective at retaining higher frequency information, even under large perturbations.

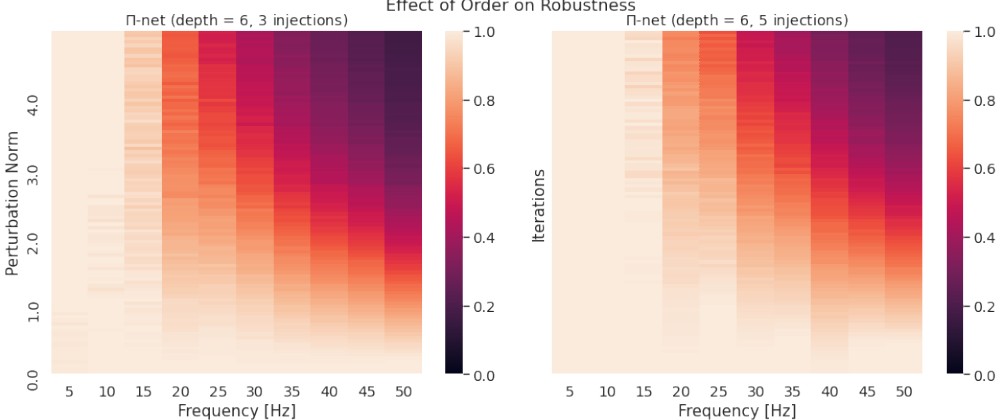

Figure 13: The heap maps present a comparison on the robustness to random parameter perturbations for two variants of Π-Nets, one with three multiplicative injection layers and the other, a higher degree polynomial with five multiplicative layers. We observe a higher degree polynomial leads to higher robustness, which is consistent with our intuition that multiplicative layers expand the solution space for learning high-frequency information.

## G EXPERIMENTS WITH IMAGES

### G.1 IMPLEMENTATION DETAILS

U-net type hourglass architectures provide the ideal inductive bias for a host of image restoration tasks in the DIP framework and for our experiments, we use the same architectural design as [33] for the standard networks. Since we require the output and input to have the same spatial dimensions, the number of downsampling blocks is equal to the upsampling blocks. We refer to this number as the 'scale' of the U-net. We provide a schematic illustration in Fig. 14.

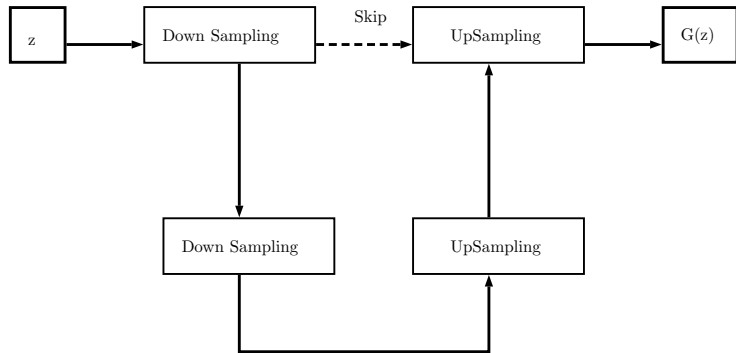

Figure 14: Illustration for the U-net with scale = 2, i.e the standard network with two upsampling/downsampling operations.

Note that the each down/upsampling block within itself contains convolutions, normalization and pooling but we abstract those details from the schematic for clarity. For the Π-network architecture, we consider a product of two polynomials model which modifies the standard network by introducing multiplicative connections. An illustrative schematic is presented in Fig. 15.

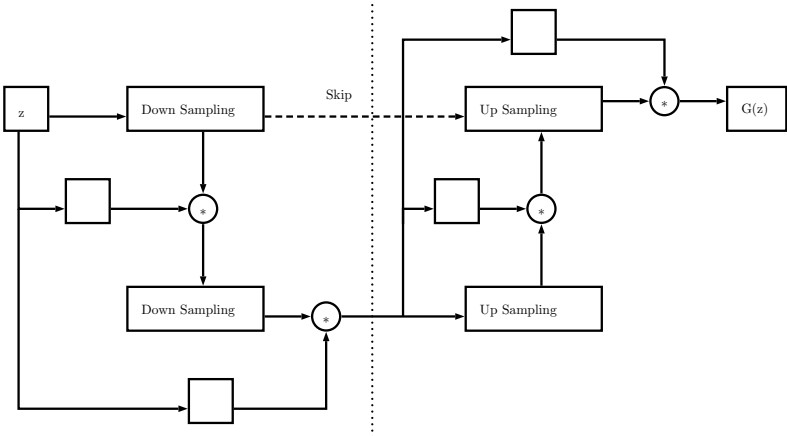

Figure 15: Adapting the 2-scale U-net for the product of two polynomials Π-network, the vertical dotted line highlights the separation between the polynomials.

**Remark**: In addition to the architecture, it is important to note another important detail about using Π-Nets. In practice, we recommend the use of two separate learning rates while training Π-Nets, wherein the learning rate for multiplicative connection parameters specifically is lower than the learning rate corresponding the parameters in the feed-forward part of the network. It leads to more stability while training.

## G.2 ADDITIONAL EXPERIMENTS

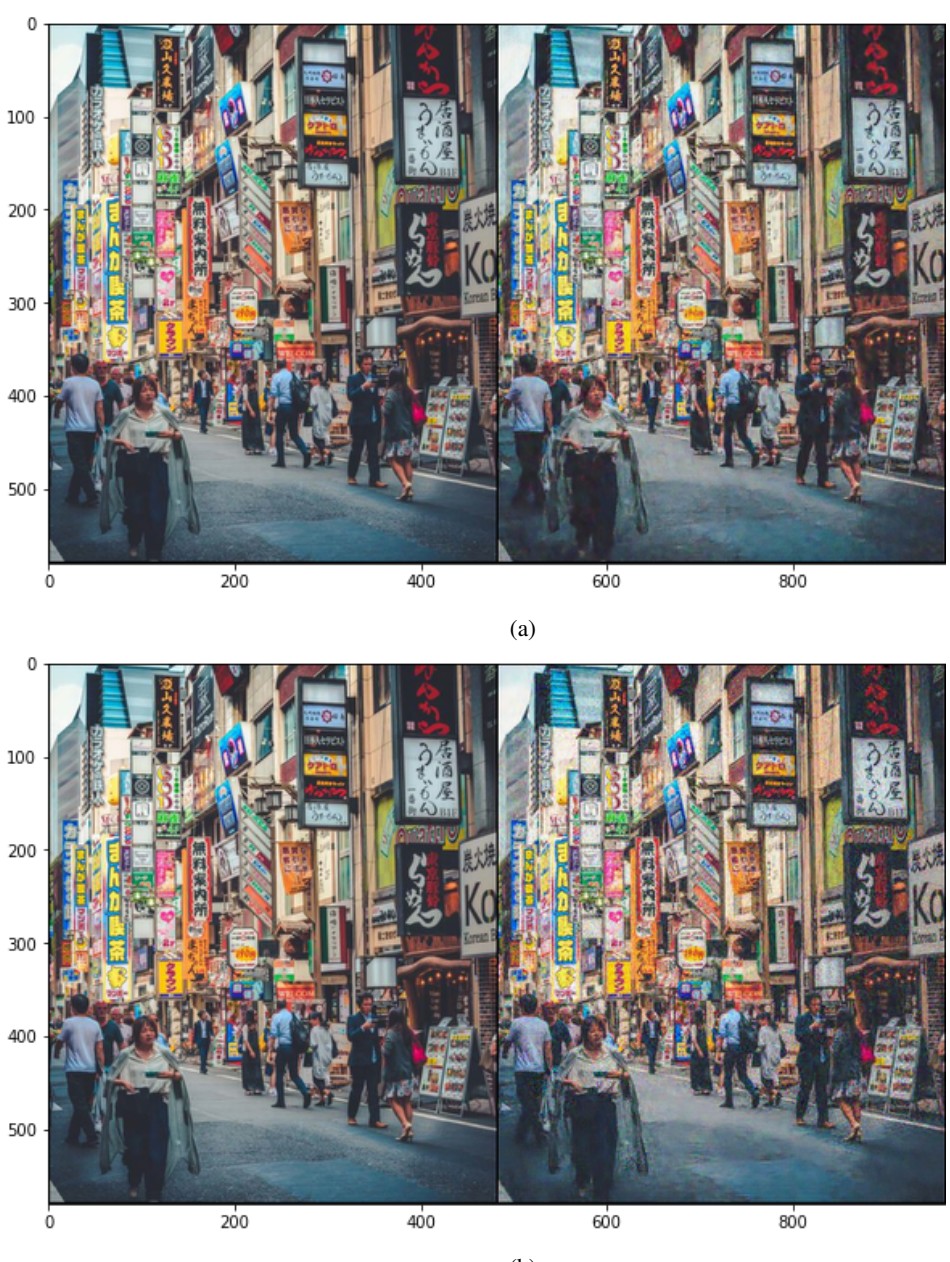

Figure 16: Visual comparison of the denoised image pertaining to the denoising experiment in 4.2. We compare a snapshot (right) of the respective network outputs after 2500 iterations against the true image (left) for (a) standard network (b) Π-Net. We visually confirm that Π-network has already begun to fit the high-frequency noise faster and to a larger extent.

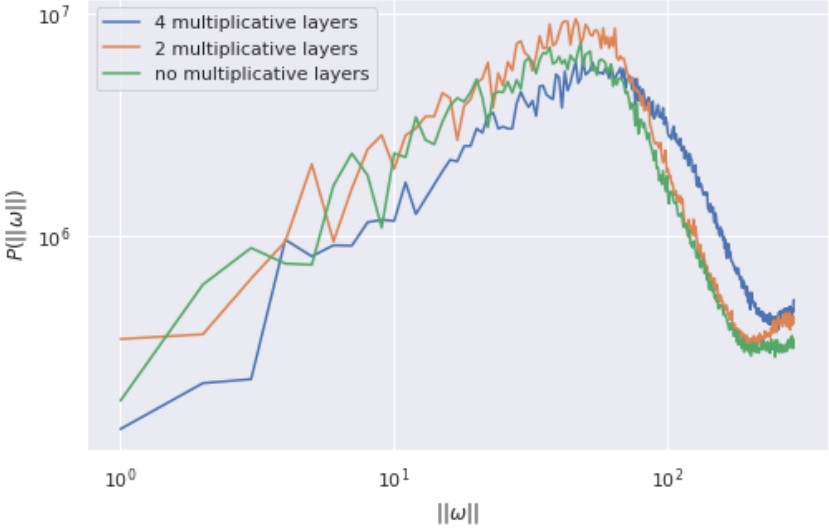

Figure 17: We visualize the deep image prior of different Π-Nets at initialization with random weights for the same input, by looking at the power spectral density of the output. With a fixed U-net architecture, introducing more multiplicative interactions in network shifts the network spectrum towards higher frequencies, which offers intuition towards understanding why multiplicative interactions speed up learning in high-frequency information.

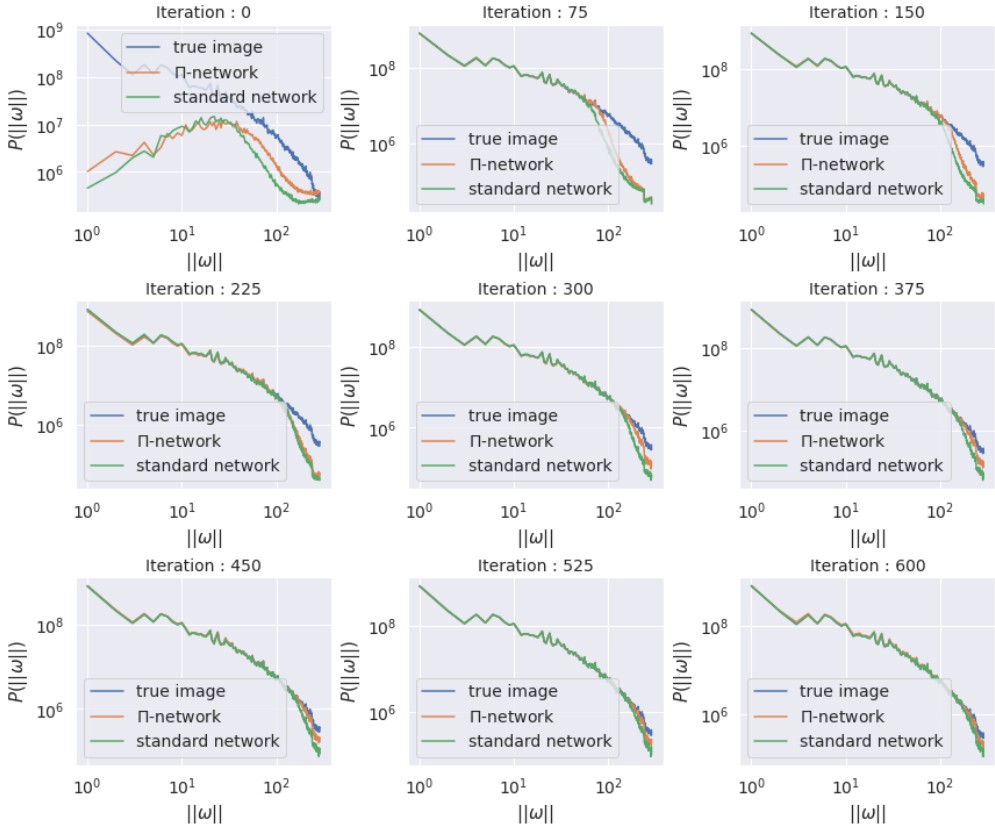

Figure 18: Comparison of the power spectral density curves over 600 iterations for standard vs Π networks (same scale) against the ground truth. We observe that the Π-Net pick up higher frequencies faster.

# H  ADDITIONAL EXPERIMENTS IN CLASSIFICATION

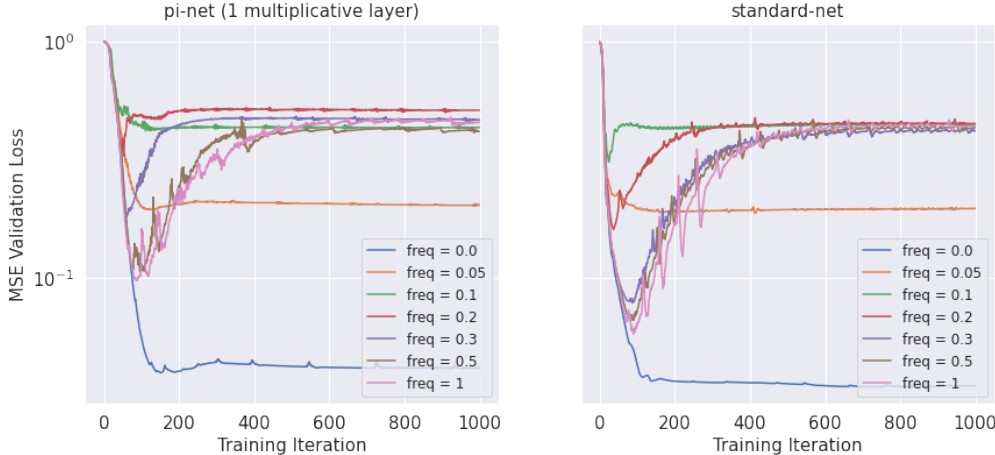

(a) Validation loss curve zoomed in for first 1000 iterations.

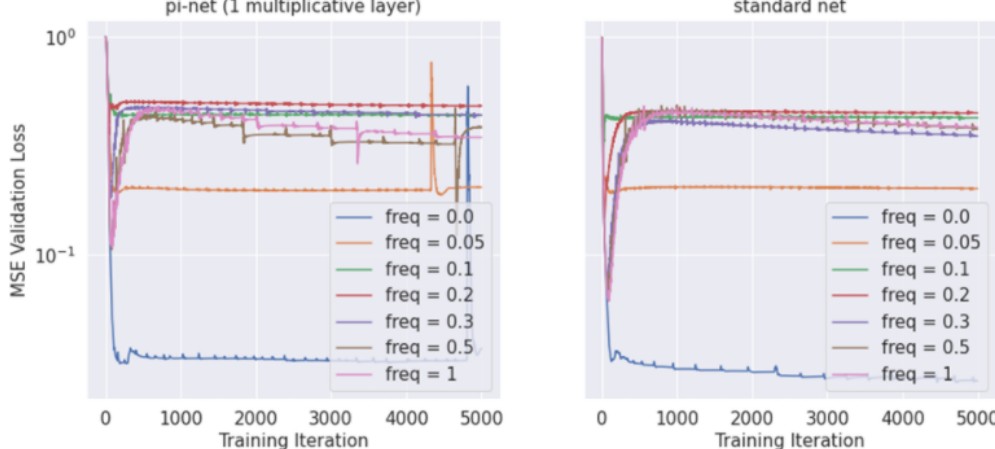

(b) Validation loss curve for 5000 iterations.

Figure 19: Validation loss curves corresponding to the classification experiment, presenting a comparison between Π-Net with one multiplicative layer and standard feedforward network. The smaller dip for Π-Net implies a tendency to pick up high frequency label noise faster.

Our results allow us to make a further overarching conclusion that Π-Nets, in addition to picking up higher frequencies w.r.t inputs faster (as demonstrated in the DIP settings), in classification settings can also pick up high frequency variations in the decision boundaries faster, thus making our general claims about the spectral bias of Π-Nets and multiplicative interactions stronger.

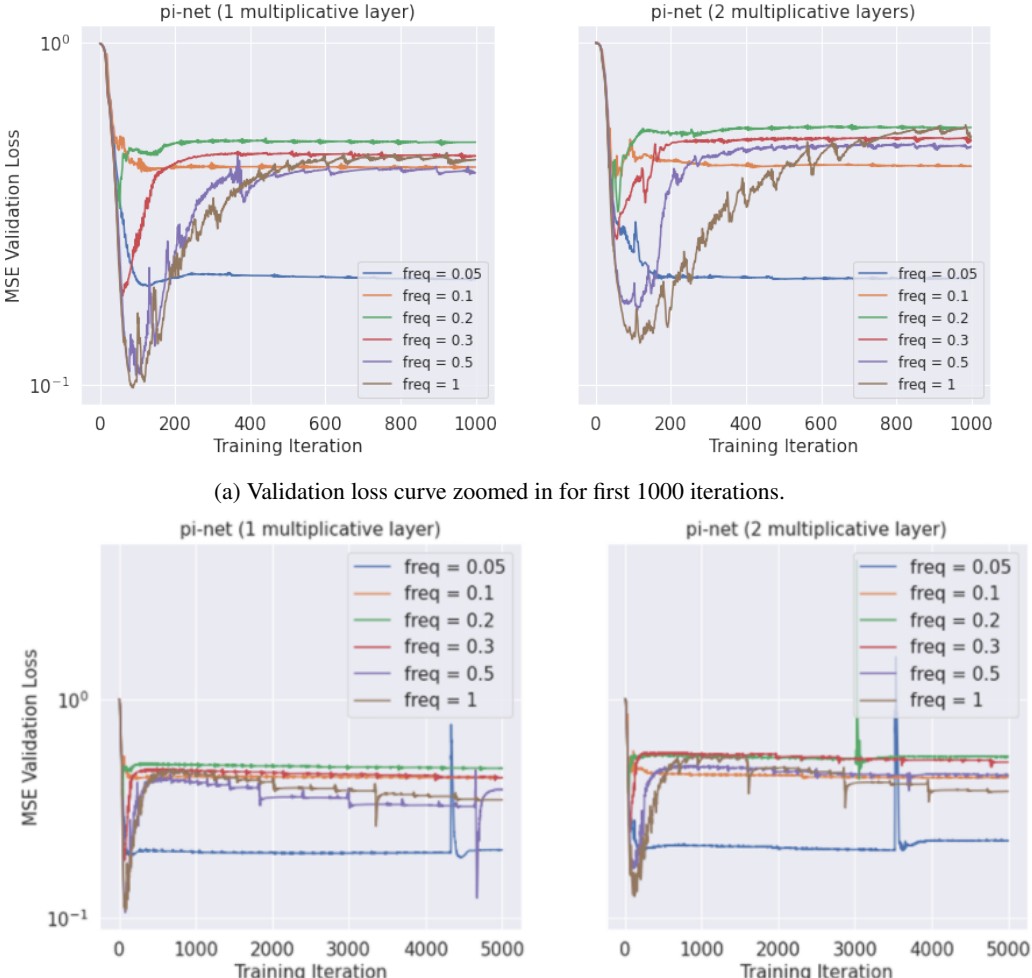

(a) Validation loss curve zoomed in for first 1000 iterations.

(b) Validation loss curve for 5000 iterations.

Figure 20: Validation loss curves corresponding to the classification experiment to observe the effect of increasing multiplicative injections. We compare the Π-Net with one multiplicative layer to Π-Net with two multiplicative layers. The validation dip reduces even further for the Π-Net with more multiplicative layers (i.e., a higher degree polynomial) indicating that more multiplicative interactions improve the network's ability to learn more complex decision boundaries (introduced by the high frequency noise).

