# OpenReview forum: "The Spectral Bias of Polynomial Neural Networks"
_ICLR.cc/2022/Conference — ICLR 2022 Poster_

### Official Review · Reviewer_FuRi · 2021-10-29

**Correctness:** 3
**Technical Novelty And Significance:** 2
**Empirical Novelty And Significance:** 3
**Recommendation:** 5
**Confidence:** 4

**Main Review:**

Strength: This paper provides an interesting viewpoint to explain why polynomial neural network performs better than feedforward neural networks in some vision tasks. The story is that PNN learns high-frequency components faster, and high-frequency components matter in these specific tasks.

Weakness:
1. Oversimplification: the model this paper analyzes can hardly be seen as a 'polynomial network'. It has two layers, which at best can express a quadratic function. For a two-layer structure, I see no necessity in forcing the $\Pi$-Net formulation. Additionally, the original PNN has no non-linear activations, while the simplified two-layer model has ReLU activations.  While I understand that the spectral analysis in previous works is performed on two-layer nets, the authors should at least explain in the paper why not analyze the full-rank polynomial network and why the ReLU activations. Currently, I feel the theoretical results for the two-layer model cannot really be applied to the PNN in practice.
2. Technical Novelty: I feel the technical novelty is lacking. The only novel contribution of this paper is to show the decay rate is slower. And to calculate the order of the eigenvalues, most efforts have been done in previous papers, except the arguments from equation (28) to equation (32). The eigenvalues of $\kappa_1$ and $\kappa_2$ are calculated in [7]; and 'larger eigenvalues imply faster learning' is proved in [9].

3. Mathematical Rigor & Presentation: it would be better to reformulate the arguments from (28) to (32), by building up inequalities between the eigenvalue and terms that are calculated. Currently, the proof looks very sloppy. When setting up the mathematical notations, it is better to keep the paper self-contained. For example, the mode-$m$ product is not introduced here. Another drawback is that the main body does not introduce why larger eigenvalues imply faster learning. This may cause a less informed audience at loss.



**Summary Of The Paper:**

This paper considers a simplified model of the polynomial neural network, and shows that the eigenvalues of the induced tangent kernel have a slower decay rate, compared with the classical two-layer network. This also means the polynomial neural networks can learn the high-frequency components faster. The empirical evidence supports this tendency.

**Summary Of The Review:**

While the topic and story of this paper are interesting, it has several drawbacks that need to be addressed. The setting is very restricted, and the theoretical results lack novelty.



========Post-rebuttal updates===========

I raise the score from 3 to 5, given that the presentation of the updated version is more rigorous and clear.

---

> ### Author Response · Authors · 2021-11-21
> **Response to Reviewer FuRi**
>
> We appreciate the feedback from the reviewer. We provide replies to their concerns below.
>
> > Oversimplification: the model this paper analyzes can hardly be seen as a 'polynomial network'. It has two layers, which at best can express a quadratic function. For a two-layer structure, I see no necessity in forcing the Pi-Net formulation.
>
> We agree with the reviewer on the comment about enforcing the $\Pi$-net structure for the 2-layer network. We have revised the paper to emphasize that this is indeed a *piecewise quadratic polynomial*. However, we believe that the formulation does motivate the structure of the multiplicative interaction layer, as well as make our empirical experiments easier to follow. Thus, we include the formulation as a primer in the main body of the work.
>
> > The original PNN has no non-linear activations, while the simplified two-layer model has ReLU activations.
>
> We respectfully disagree with the reviewer; the implementation in $\Pi$-nets includes both the case with ReLU activation functions and without. For instance, the experiments on face recognition use activation functions, as can be seen in the public implementation of the $\Pi$-net code [F].
>
> > While I understand that the spectral analysis in previous works is performed on two-layer nets, the authors should at least explain in the paper why not analyze the full-rank polynomial network and why the ReLU activations.
>
> We thank the reviewer for pointing this out, we have revised the relevant parts in the paper to explain our rationale behind picking this setting.
>
> > Currently, I feel the theoretical results for the two-layer model cannot really be applied to the PNN in practice.
>
> We agree that our theoretical analysis cannot be directly extended to higher degree polynomials in general. At the same time, we believe that it is an **important step to understand the effect of multiplicative interactions**. Additionally, our experiments offer *ample evidence* to bridge this gap to some extent and demonstrate that this spectral bias indeed does manifest in practice beyond the NTK regime, in both fully connected and deep convolutional networks.
>
> > Technical Novelty: I feel the technical novelty is lacking. The only novel contribution of this paper is to show the decay rate is slower. And to calculate the order of the eigenvalues, most efforts have been done in previous papers, except the arguments from equation (28) to equation (32). The eigenvalues of and are calculated in [7]; and 'larger eigenvalues imply faster learning' is proved in [9].
>
> We agree that our paper does not introduce new tools or proof techniques from the point of view of deep learning theory. However, we strongly believe that the application of existing tools to further the understanding of a strongly motivated problem has value in and of itself and we believe this is where the strengths of our work lie. Even though the NTK regime has been used in the past for studying the order of the eigenvalues, we are not aware of any previous study for networks that include multiplicative interactions. Consequently, **the proofs with this class of functions have not been conducted before**.
>
> Starting from a clear motivation (agreed upon by reviewers TQnp, KnZp, cEcf), we derive the $\Pi$-net kernel and prove that it results in faster learning of higher-frequencies which we believe is an important step towards analyzing network structures in the NTK regime that deviate from standard feedforward networks.
>
> > Mathematical Rigor & Presentation: it would be better to reformulate the arguments from (28) to (32), by building up inequalities between the eigenvalue and terms that are calculated. Currently, the proof looks very sloppy. When setting up the mathematical notations, it is better to keep the paper self-contained. For example, the mode- product is not introduced here. Another drawback is that the main body does not introduce why larger eigenvalues imply faster learning. This may cause a less informed audience at loss.
>
> We have significantly revised the presentation of the proof and also included the required notation. Due to lack of space, we have included a short explanation of how larger eigenvalues lead to faster learning in the appendix (pg. 16). We are thankful to the reviewer for pointing out specific changes in the manuscript and we believe that the current version is indeed improved.
>
> References
>
> [F] [https://github.com/grigorisg9gr/polynomial_nets/tree/master/face_recognition](https://github.com/grigorisg9gr/polynomial_nets/tree/master/face_recognition)

---

> > ### Author Response · Authors · 2021-11-29
> > **Did our response cover the concerns of the reviewer?**
> >
> > Dear reviewer FuRi,
> >
> > We are thankful for your efforts so far. Since the rebuttal period is closing soon, we would like to inquire whether the reviewer has any remaining questions on the setting, given our clarification on the use of activation functions and other technical choices. In addition, we would like to inquire whether the improvements in the presentation of the paper and proposed improvements, regarding including the recursive $\Pi$-Net formulation with ReLU activations in the main body and the new experiment in [image classification](https://openreview.net/forum?id=P7FLfMLTSEX&noteId=EWOwg3055aw), have improved your opinion of our work.

---

> > > ### Comment · Reviewer_FuRi · 2021-11-29
> > > **Increasing to score 5**
> > >
> > > Dear authors,
> > >
> > > Thanks for your responses and clarification. Most of my concerns are addressed. I went over the new submission and felt satisfied with the presentation now.
> > >
> > > Still, I have to say that the theoretical part 1) follows the existing framework and toolset, and 2) works on a rather simplified and impractical model. Either of these drawbacks alone shouldn't stand for rejection, but with both of them, I cannot recommend acceptance. Therefore, I will raise the score to 5.

---

> > > > ### Author Response · Authors · 2021-11-30
> > > > **Response to response of reviewer FuRi**
> > > >
> > > > We are thankful for the timely response and for appreciating our effort.
> > > >
> > > > We agree that we follow an existing framework and toolset, but we see using the established models for a new problem as a strength. That is, the observation we make on the effect of adding a multiplicative layer to neural network has not been made before and we use these tools to relate and compare in a straight-forward way with results on standard neural networks.
> > > >
> > > > Regarding the remark on using a simplified setting, we concede to the limitation that our theoretical results do not directly extend to higher degree polynomials. However, we do make a consistent effort to bridge this gap in several experiments that start from synthetic settings and progressively increase the dimensionality/complexity of the setup, including polynomial networks of higher degrees. Indeed, the empirical results strongly verify this spectral bias, and clearly align with our theoretical intuition from networks near initialization in the NTK setting. Thus, we believe that our theory along with our experiments, together form a convincing story that demonstrates new, relevant, and impactful knowledge which will be valuable to the ICLR community, and open up multiple avenues for future research.
> > > >
> > > > Finally, if the reviewer has any concrete ideas on how to proceed towards extending the theoretical setup, we would be grateful for the feedback and glad to include them in our future steps.

---

### Official Review · Reviewer_cEcf · 2021-11-02

**Correctness:** 4
**Technical Novelty And Significance:** 2
**Empirical Novelty And Significance:** 3
**Recommendation:** 6
**Confidence:** 3

**Main Review:**

Overall I think this paper is well organized and well written. The proof of this paper seems solid. The main theoretical contributions of this paper are Theorem 1 and 2. Theorem 1 gives the kernel limit of two-layer $\Pi$-net. And Theorem 2 computes the eigenvalues of the linear operator. Given the techniques used in reference [9], the proof of Theorem 1 and 2 are quite strict forward.  The empirical results are quite clear and well supports the main theorem. It's is quite interesting that $\mu_{k} = \Omega(k^{-d/2 -2 })$ rather than $\mu_{k} = \Omega(k^{-d - 1})$.  What will happen if we increase the number of layers or the degree?

**Summary Of The Paper:**

The paper gives an analysis of polynomial networks in the NTK regime. They show a theoretical speed-up in learning higher frequencies when using polynomial networks. They also have some experiments that verify these properties.

**Summary Of The Review:**

This paper is well organized and well written.  The proof is solid, and the empirical results support its claim. However, given the previous paper the, proof techniques are not novel. Therefore, I would only suggest a weak acceptance.

---

> ### Author Response · Authors · 2021-11-21
> **Response to Reviewer cEcf**
>
> We are thankful to the reviewer for the feedback and the appreciation of our work. We address their concerns below.
>
> > Overall I think this paper is well organized and well written. The proof of this paper seems solid. The main theoretical contributions of this paper are Theorem 1 and 2. Theorem 1 gives the kernel limit of two-layer -net. And Theorem 2 computes the eigenvalues of the linear operator. Given the techniques used in reference [9], the proof of Theorem 1 and 2 are quite straightforward. The empirical results are quite clear and well support the main theorem. It's quite interesting . What will happen if we increase the number of layers or the degree?
>
> We thank the reviewer for the positive review. Indeed, we believe that extending our theoretical analysis to more general higher degree polynomials is an important future direction, either within the NTK regime or using other tools.
>
> One interesting phenomenon that we noted in our experiments was that the performance of the multiplicative layers strongly varies based on the starting layer (the input to the multiplicative layer) and the end point (the particular output layer to which the hadamard product is applied to) in the network. We speculate that this observation is intimately tied to understanding how representations for different layers vary as one moves deeper through the layers of the neural network. Thus, leveraging recent works on how optimal representations emerge during training [E] for the optimal design of multiplicative injection layers is an exciting avenue of research on the empirical front.
>
> Observing the performance of polynomial networks from the generalization point of view is another important direction. We mention this in the sec.5 (discussion). The idea is that multiplicative interactions enhance the spectral bias for higher frequencies i.e. towards learning higher “complexity” functions. Therefore, it is imperative to study if and how multiplicative interactions affect generalization, since it is believed that standard neural networks generalize well partly due to their spectral bias towards low complexity functions.
>
> References
>
> [E]  Michael Kleinman, Alessandro Achille, Daksh Idnani, Jonathan Kao: Usable Information and Evolution of Optimal Representations During Training. ICLR 2021

---

> > ### Author Response · Authors · 2021-11-29
> > **Are there any remaining questions from the reviewer?**
> >
> > Dear reviewer cEcf,
> >
> > We are thankful for your constructive feedback. Regarding the technical proof aspects, we agree with the reviewer that our paper does not introduce novel techniques from a deep learning theory perspective. However, we make a surprising finding that helps motivate an explanation towards observations already made in the literature (such as state-of-the-art generative models being polynomials or multiplicative interactions improving the inductive bias for certain tasks).
> >
> > Since this is the last day of the discussion period, we would like to check if the reviewer has been covered by our explanation, improvements made to the manuscript based on suggestions as well as new empirical evidence in [image classification](https://openreview.net/forum?id=P7FLfMLTSEX&noteId=EWOwg3055aw) and if there are any remaining questions that we could address. We are thankful for your feedback.

---

### Official Review · Reviewer_nFMY · 2021-11-03

**Correctness:** 3
**Technical Novelty And Significance:** 3
**Empirical Novelty And Significance:** 2
**Recommendation:** 6
**Confidence:** 2

**Main Review:**

Note that the network (7) for which NTK kernel was calculated and spectral analysis is made, is not a Π-Net (because it includes ReLU as sigma). So, why can we apply the spectral analysis to Π-Net architecture?

Experiments seem to support the main claim, though fig 5 shows that the difference from standard network is not so sharp. Eventually, it seems that the standard network is also able to learn the high frequency. Also, experimental setups are taken from well-known papers.



**Summary Of The Paper:**

The paper is dedicated to the study of spectral bias in polynomial neural networks. For feed-forward neural networks with ReLU activation function it is well-known that learning high-frequency terms is made slower (Rahaman et al.).

Authors claim that polynomial networks are able to learn both low frequency and high-frequency terms almost equally fast. Theory, based on spectral analysis of an integral operator associated with the NTK kernel, is provided.  Theoretical approach was taken from arXiv: 1905.12173, but for NTK kernel of the network with an additional multiplicative interaction layer.
Then, experiments that support claims are provided.
It is claimed that multiplicative interactions in the architecture of a NN may be the reason for an ability to learn high frequencies.


**Summary Of The Review:**

The narrative is well structured and easy to read. The main claim is partially supported. Weak aspects of Π-nets are not discussed. I would assess that it is a borderline paper.

---

> ### Author Response · Authors · 2021-11-21
> **Response to Reviewer nFMY**
>
> We appreciate the feedback from the reviewer. We provide replies to their concerns below.
>
>  > Note that the network (7) for which NTK kernel was calculated and spectral analysis is made, is not a Π-Net (because it includes ReLU as sigma). So, why can we apply the spectral analysis to Π-Net architecture?
>
> We hope we were able to justify our choice of setting for the theoretical analysis in the first point of our common response.
>
>  > Experiments seem to support the main claim, though fig 5 shows that the difference from standard network is not so sharp. Eventually, it seems that the standard network is also able to learn the high frequency. Also, experimental setups are taken from well-known papers.
>
> We would like to point out that with sufficiently overparameterized networks, we observe that standard networks eventually pick up high frequency information but they do so slower than polynomial networks, thus leading to a computational speed-up. However, in settings with restricted capacity, we can expect polynomial networks with multiplicative injections to outperform standard networks.
>
> Regarding our experimental setups being adopted from existing works, we believe that to be one of the strengths of our work since the setups have already been validated by the research community, which makes our claims much more believable.
>
> > The narrative is well structured and easy to read. The main claim is partially supported. Weak aspects of Π-nets are not discussed. I would assess that it is a borderline paper.
>
> We have added the relevant discussion about $\Pi$-Nets, including a note on how this spectral bias can affect generalization, and other potential directions of future work more explicitly in the discussion section.

---

> > ### Author Response · Authors · 2021-11-29
> > **Did our response cover the questions of the reviewer?**
> >
> > Since this is the last day of the discussion period, we would like to check if the reviewer has been covered by our explanation and the updates on the paper (e.g. to discuss the weak aspects of our analysis and the new experiment in [image classification](https://openreview.net/forum?id=P7FLfMLTSEX&noteId=EWOwg3055aw)) or if there are any remaining questions that we could address. We are thankful for their feedback.

---

### Official Review · Reviewer_KnZp · 2021-11-06

**Correctness:** 4
**Technical Novelty And Significance:** 3
**Empirical Novelty And Significance:** 2
**Recommendation:** 8
**Confidence:** 3

**Main Review:**

Main strengths of the paper
1. Ability to control the spectral bias of neural networks is an important and mostly open question, at least for networks that operate on high-dimensional inputs. This paper offers a valuable step in this direction by showing that multiplicative connections reduce the low-frequency bias of neural networks.
2. The paper is clearly written.

Main limitations of the paper
1. The theoretical contribution is limited to a very restrictive model: 2 layers, no nonlinear activation function, no bias term, and many more data points than parameters. Further, the analysis is in the NTK regime, which assumes that width tends towards infinity and learning rate tends towards zero.
2. Although the experiments do test violating the 2-layer assumption and the NTK regime assumptions, they remain restricted to very low dimensional problems (1D synthetic functions and 2D images) where, to the best of my knowledge, a neural network would not be the go-to method in real applications.
3. The related work section is very sparse. This should include prior work on spectral bias of ReLU networks (some of which is included in the introduction, but not all--for example, https://arxiv.org/abs/1906.00425 and https://arxiv.org/abs/2003.04560). It should also include other methods that affect spectral bias, for example https://arxiv.org/abs/2006.10739.

Low-level suggestions
1. Claim 2 could be more specific. Upon a first reading of “standard inverse problems in imaging” I expected some 3D imaging or CT or MRI experiments, but this is actually referring to 2D image denoising.
2. Equation (1) uses the “mode-m vector product”. A quick search showed me the definition of this operation, but it would be easier for the reader if this were defined in the paper (perhaps with examples in the case of vectors and matrices).
3. Equation (2) could be visualized with a simple diagram of the 2-layer pi-net you analyze.
4. It would be helpful for the last sentence of section 3.3 to mention the Schur product theorem as justification.
5. It would be helpful to more clearly delineate the difference between the first and second experiments in section 4.1. As far as I can tell the difference is that the first uses large width and low learning rate and the second uses a smaller width and larger learning rate (to violate NTK assumptions), but these should be stated explicitly.
6. Figure 2 is not very convincing--actually I find Figure 11 in the appendix to be a more compelling comparison, both visually and in terms of the relevance of the comparison.


**Summary Of The Paper:**

The paper presents a theoretical analysis of the RKHS of the NTK of two-layer polynomial neural networks without bias terms or nonlinear activations. The primary difference between these networks and standard ReLU networks is the presence of (elementwise) multiplicative skip connections, which the authors show (both in the NTK analysis and experiments) improves the convergence rate for higher frequency target functions.

**Summary Of The Review:**

The paper offers an interesting approach to reduce the spectral bias of neural networks by introducing multiplicative connections, and supports this claim with NTK analysis of shallow networks and experimental evidence on low-dimensional synthetic and real data. Although both the theoretical and experimental contributions are rather restricted, I hope they will serve as useful building blocks for subsequent research on more realistic models and data.

---

> ### Author Response · Authors · 2021-11-21
> **Response to Reviewer KnZp**
>
> We thank the reviewer for their feedback. We address below their concerns.
>
> > The theoretical contribution is limited to a very restrictive model: 2 layers, no nonlinear activation function, no bias term, and many more data points than parameters. Further, the analysis is in the NTK regime, which assumes that width tends towards infinity and learning rate tends towards zero.
>
> We hope we were able to justify our choice of setting for the theoretical analysis in the first point of our common response.
>
> > Although the experiments do test violating the 2-layer assumption and the NTK regime assumptions, they remain restricted to very low dimensional problems (1D synthetic functions and 2D images) where, to the best of my knowledge, a neural network would not be the go-to method in real applications.
>
> We respectfully disagree with the reviewer. The experiment we conduct in the Deep Image Prior setting uses an image that is of 480 x 600 resolution (now referenced in the paper). We believe that one of the main reasons behind the resurgence of neural networks back in 2012 was their success in image classification [L]. And indeed many of the successes since have been demonstrated on 2D images [J, K].
>
> However, we agree with the reviewer that the goal of this work is not to demonstrate state-of-the-art behavior in the latest benchmarks, but rather to explain a phenomenon. To that end, we have considered a number of experiments with increasing dimensionality and complexity to complement our theoretical motivation on NTK regime. Conducting large-scale experiments to further build on the success of polynomial networks is a future research direction, where we expect the insights from our work to be useful.
>
> > The related work section is very sparse. This should include prior work on spectral bias of ReLU networks (some of which is included in the introduction, but not all--for example, https://arxiv.org/abs/1906.00425 and https://arxiv.org/abs/2003.04560). It should also include other methods that affect spectral bias, for example https://arxiv.org/abs/2006.10739.
>
> We thank the reviewer for the suggestions and providing additional references. We have updated the related work section to better reflect the state of current work and our contributions. We will update here with experimental results during the next week.
>
> > Low Level Suggestions
>
> We have updated the manuscript as per the provided suggestions and we are thankful for their keen observations.
>
> References
>
> [J] : Syed Muhammad Arsalan Bashir, Yi Wang, Mahrukh Khan and Yilong Niu: A Comprehensive Review of Deep Learning-based Single Image Super-resolution, 2021 - [https://arxiv.org/abs/2102.09351](https://arxiv.org/abs/2102.09351)
>
> [K] Seungjun Nah, Sanghyun Son, Suyoung Lee, Radu Timofte, Kyoung Mu Lee: NTIRE 2021 Challenge on Image Deblurring, 2021 - [https://arxiv.org/abs/2104.14854](https://arxiv.org/abs/2104.14854)
>
> [L] Alex Krizhevsky, Ilya Sutskever, Geoffrey E. Hinton: ImageNet Classification with Deep Convolutional Neural Networks, NeurIPS 2012

---

> > ### Comment · Reviewer_KnZp · 2021-11-29
> > **Response to authors**
> >
> > Thank you for your response and for updating the paper. Regarding low vs high dimensional problems, this is somewhat of a semantic distinction: a problem like denoising or super-resolution is fundamentally trying to predict a function (the image colors) in two dimensions (x and y, over the image pixels), hence I refer to this as a low-dimensional problem. This is in contrast to something like image classification, in which the goal is to predict a function (class membership) over the space of all images, a very high-dimensional problem. Traditional approaches to these "low-resolution" problems use techniques from digital signal processing, which use the prior that the true image must be bandlimited. More recent neural net approaches to these problems use datasets of many images to learn more specific priors that can produce sharper results than traditional methods. Whether or not this use of many images to inform a prior makes these neural methods "high-dimensional" is a semantic distinction that we don't need to dwell on.

---

> > > ### Author Response · Authors · 2021-11-29
> > > **Response to Response to authors (Reviewer KnZp)**
> > >
> > > We thank the reviewer for the clarification. We hope that our new experiment with MNIST, while admittedly a simplification of a multi-class classification problem, offers a step towards understanding/observing the spectral bias in high-dimensional problems.

---

### Official Review · Reviewer_TQnp · 2021-11-07

**Correctness:** 3
**Technical Novelty And Significance:** 3
**Empirical Novelty And Significance:** 2
**Recommendation:** 6
**Confidence:** 3

**Main Review:**

The paper is well written and explains the problem as well as the approach clearly. I appreciate the authors for spending their time polishing the writing, providing clean theorems, and reviewing the literature.

I also think that the problem presented here is meaningful and worth further studying. The fact that PNNs well learn high-frequency signals while general NNs don't indicate that some unique attributes of PNNs enable the model to retain a certain level of signal-to-noise ratio. The existing and current mathematical analyses on this open problem have shone a light on practical applications. Personally

One of the main concerns is that the paper studies only two-layer PNNs, which is restrictive. The authors stated in the abstract that "we expect our analysis to provide novel insights into designing architectures and learning frameworks..." But it seems to me that two-layer NNs may demonstrate significant behavior compared to DNNs. For example, I am not sure if the fairness of two-layer NNs extends to deeper models. Also, the analysis leverages an infinite-width assumption, far from being practical.

Another concern is regarding the experiments. The experiments considered the performance of NNs with more than two layers. Though the previous theoretical analysis does not apply to this setting, more layers make the scenarios more practical. But given the scale of the experiments, I think it might be reasonable to consider other supervised learning methods here. Alternatively, as the setting already violates the two-layer assumption, maybe it is appropriate to consider practical applications with high-dimensional and large datasets.

Finally, I suggest the authors provide a better discussion in the related work section. Currently, it contains only two paragraphs, not delivering sufficient insights. In particular, it would be good to add comparisons (theories & techniques & proofs) between the essential references and the submission, i.e., what makes the paper special?


**Summary Of The Paper:**

The paper studies ReLU-activated two-layer neural networks. The main contribution is that unlike traditional deep neural networks (DNNs), this type of polynomial neural network (PNN) also learns high-frequency information quickly. This claim partially verifies why PNNs are suitable for tasks like face recognition, where high-frequency components are crucial for performance.

**Summary Of The Review:**

I like the writing quality and the problem. While it is common to start with simplified models for theoretical analysis, I expect a more profound theorem addressing broader PNN families given the existing work. I appreciate the authors for discussing the prior work and performing experimental comparisons. But I think both aspects are subject to further improvements, i.e., the experiments should emphasize practicality, and the literature review could help differentiate the paper's contributions.

---

> ### Author Response · Authors · 2021-11-21
> **Response to Reviewer TQnp**
>
> We thank the reviewer for the comments. We address below their concerns.
>
> > One of the main concerns is that the paper studies only two-layer PNNs, which is restrictive. The authors stated in the abstract that "we expect our analysis to provide novel insights into designing architectures and learning frameworks..." But it seems to me that two-layer NNs may demonstrate significant behavior compared to DNNs. For example, I am not sure if the fairness of two-layer NNs extends to deeper models. Also, the analysis leverages an infinite-width assumption, far from being practical.
>
> We hope we were able to justify our choice of setting for the theoretical analysis in the first point of our common response.
>
> > Another concern is regarding the experiments. The experiments considered the performance of NNs with more than two layers. Though the previous theoretical analysis does not apply to this setting, more layers make the scenarios more practical. But given the scale of the experiments, I think it might be reasonable to consider other supervised learning methods here. Alternatively, as the setting already violates the two-layer assumption, maybe it is appropriate to consider practical applications with high-dimensional and large datasets.
>
> We want to stress that the goal of this work is not to demonstrate state-of-the-art behavior in the latest benchmarks, but rather to understand a phenomenon pertaining to multiplicative interactions. To that end, we consider a number of experiments with increasing dimensionality and complexity to complement our theoretical motivation from the NTK regime.
>
> We agree with the reviewer’s sentiment that having similarly informative results in state-of-art benchmarks would strengthen our case, but at the same time we want to point out that it becomes increasingly harder to disentangle the spectral effects in more complicated settings due to other confounding factors. For instance, attempting a similar spectral bias study with the GAN framework for image generation would require disentangling effects and restrictions due to learning rates or the capacity of the discriminator (a discriminator might simply ignore information in higher frequencies during early stages of training, thus affecting the generator). The difficulty in isolating these effects can be seen in concurrent studies on spectral bias, that attempt to extend artificial label noise experiments from Rahaman et. al on classification tasks, but with only limited success [H]. For this reason, we consider simpler experiments to deliver our message concisely.
>
> We do believe that carefully designing settings for conducting large scale experiments is an important future step, based on the intuitions provided both theoretically and with various experiments in this work, and we have introduced a sentence in sec.5 (discussion) to emphasize the same.
>
> > Finally, I suggest the authors provide a better discussion in the related work section. Currently, it contains only two paragraphs, not delivering sufficient insights. In particular, it would be good to add comparisons (theories & techniques & proofs) between the essential references and the submission, i.e., what makes the paper special?
>
> We thank the reviewer for the suggestions and have updated the related work section to better reflect the state of current work and our contributions. We will update here with the experimental results during the next week.
>
> References
>
> [H]  Spectral Bias in Practice: the Role of Function Frequency in Generalization (ICLR 2022, under review) - https://openreview.net/forum?id=e-IkMkna5uJ

---

> > ### Author Response · Authors · 2021-11-29
> > **Are there any remaining questions from the reviewer?**
> >
> > Since this is the last day of the discussion period, we would like to check if the reviewer has been covered by our explanations and the newly added experiment on [image classification](https://openreview.net/forum?id=P7FLfMLTSEX&noteId=EWOwg3055aw) or if there are any remaining questions that we could address. We are thankful for their feedback.

---

### Author Response · Authors · 2021-11-21
**Common Response to the reviewers**

We thank the reviewers for their constructive feedback. We are pleased to note the motivation of our work resonates with the reviewers (Reviewers TQnp, KnZp, cEcf), the writing and narrative is easy to follow (Reviewers TQnp, KnZp, nFMY, cEcf). We address two common concerns below:

1. A remark raised was that the theoretical results for the 2-layer $\Pi$-networks in the NTK regime are restrictive. We explain the rationale behind choosing this setting:
  * To our knowledge, the Neural Tangent Kernel (NTK) has been a significant tool in the last few years for the theoretical analysis of deep networks. Even though there is of course still a gap between the NTK theory and practice, it has become one of the most standard tools for building upon our theoretical understanding of deep networks [A, G].  We thus believe it to be a good starting point for our analysis for polynomial networks.
  * The rationale for limiting our analysis to 2-layer polynomial networks is twofold: the first reason is a known limitation of the NTK regime [I], in that the error bounds (i.e. the error in approximating the deep net using the network at initialization for a fixed width) degrade with depth. Additionally, recent works suggest that within the NTK regime, _deep networks have essentially the same approximation properties (eigenvalue decay) as their shallow 2-layer counterparts_ [B].
  * The second reason relates to the design of polynomials, where the exact structure of the multiplicative injections can drastically alter the network kernel. As an example, consider an N-layer ReLU feedforward network with just one multiplicative injection. By simply changing the placement (start and end point) of the multiplicative connection, we obtain different limiting kernels. For deeper networks, the placement of the multiplicative layer ties into the larger question of “What is the best polynomial for learning?”, which in itself is an important direction for future work. Thus, to limit the effect of this confounding factor, we consider the simplest 2-layer $\Pi$-Net with one multiplicative layer.
  * The added advantage of this choice is that it allows us to directly contrast against prior results on standard 2-layer networks, thereby gauging the effect of the multiplicative layer, which aside from polynomial networks, also appear in other popular architectures and frameworks. It also explains how multiplicative layers enhance the inductive bias of neural networks [C] from the spectral bias perspective.
  * Next, it is important to point out the reason for considering non-linear activations (ReLU) for this analysis. The reviewers will note that without non-linear activations, the NTK for a finite depth $\Pi$-net of degree $n$ will be a polynomial kernel of degree n and thus, will have eigenvalues corresponding to harmonics of higher degrees equal to 0. Consequently under the current setup, $\Pi$_-Nets without activations will not approximate higher frequencies even with infinite width._ In fact in the original paper [M], $\Pi$-Nets achieve state-of-the-art results for image generation and classification only with activation functions and we thus believe it to be a fair inclusion.
  * Finally, we should note that while inclusion of an explicit bias term (usually via appending an extra dimension to the input and renormalizing) speeds up learning for odd harmonics, it does not alter the asymptotic decay rate of $O(k^{-d})$ [D]. Therefore, we choose to exclude it from our analysis for simplicity.

    *We are thankful to the reviewers for raising this point*. The updated manuscript provides a better understanding of our choice of model as well as the limitations of our NTK analysis, since we agree that more work is needed to extend our analysis to higher degree polynomials.

2. Upon the request of the reviewers, we have extended the related work (sec. 2) to include additional references. In addition, we have emphasized the differences of the related works from our contributions. We appreciate the remark on the related work from the reviewers and pointing out the relevant references.

We believe that the revised manuscript is strengthened from the further discussions and clarifications. We *kindly ask the reviewers to share any remaining questions*.

---

> ### Author Response · Authors · 2021-11-21
> **References in Common Response**
>
> References
>
> [A] Quynh Nguyen, Marco Mondelli, Guido Montufar: Tight Bounds on the Smallest Eigenvalue of the Neural Tangent Kernel for Deep ReLU Networks. ICML 2021
>
> [B]  Alberto Bietti, Francis Bach: Deep Equals Shallow for ReLU Networks in Kernel Regimes.  ICLR 2021
>
> [C] Jayakumar et al.: Multiplicative Interactions and Where to Find Them. ICLR 2020
>
> [D] Ronen Basri, David W. Jacobs, Yoni Kasten, Shira Kritchman: The Convergence Rate of Neural Networks for Learned Functions of Different Frequencies. NeurIPS 2019
>
> [G] Amir Zandieh, Insu Han, Haim Avron, Neta Shoham, Chaewon Kim, Jinwoo Shin: Scaling Neural Tangent Kernels via Sketching and Random Features. NeurIPS 2021
>
> [I] Deep Learning Theory (CS540) - [Deep learning theory lecture notes (illinois.edu)](https://mjt.cs.illinois.edu/dlt/index.pdf)
>
> [M] Grigorios G. Chrysos, Stylianos Moschoglou, Giorgos Bouritsas, Yannis Panagakis, Jiankang Deng, Stefanos Zafeiriou: P-nets: Deep Polynomial Neural Networks. CVPR 2020

---

> ### Comment · Reviewer_KnZp · 2021-11-29
> **Followup question**
>
> Thank you for justifying the choice of regime used in the theoretical analysis. I’m still a little bit confused about the nonlinear activations: specifically, in section 3.1 the pi-net formulation does not include nonlinear activations, whereas in section 3.3 activations are included. Is there a reason to use different formulations in different sections? If the nonlinear activations are important, and your analysis does include them, then why not include them in the definition of a pi-net?

---

> > ### Author Response · Authors · 2021-11-29
> > **Response to Followup question from Reviewer KnZp**
> >
> > Our theoretical analysis indeed relies on activation functions, which is why we include the activation functions in our experiments (i.e., in sec. 3.3, sec.4 and the newly added classification experiment). Initially, we included the original formulation of $\Pi$-nets in sec. 3.1 for making the manuscript self-contained, however we acknowledge that adding the activation functions in eq. 2 would give a clearer understanding to the reader of the used formula. To that end, we moved the current content of sec. 3.1 (with the pure polynomial expansion without activation functions) to the Appendix and **we updated sec. 3.1 with a recursive formulation of $\Pi$-nets with RELU activation functions** to have a direct correspondence with the network used in the numerical results. We will upload the updated manuscript when we are allowed to do so.

---

### Author Response · Authors · 2021-11-29
**New Experiment in Classification settings**

**_*tl-dr:* We extend an experiment based on the spectral bias on neural networks [A] and verify that indeed high-frequency noise affects the validation accuracy of polynomial networks in a classification setting._**

We conduct an experiment to supplement our previous results on the DIP setting. Our goal is to provide complementary results in a classification task as requested by the reviewers. Specifically, we evaluate the robustness of the networks to label noise, following the setting of Rahaman et al. [A].

The experiment considers the fully connected 6-layer deep 256-unit wide network (as in sec. 4.1 of our paper) on MNIST, restricting the problem to binary classification on images of classes “3” and “8”, following the same setup as described in [A, sec. 3.2]. From the results in [A], the following observations were made for **[standard neural networks] ([https://openreview.net/forum?id=r1gR2sC9FX&noteId=H1xjnEG4eE](https://openreview.net/forum?id=r1gR2sC9FX&noteId=H1xjnEG4eE)**):

(a) For a fixed amplitude, adding a low-frequency noise signal to labels degrades the generalization performance (the difference between training and validation losses) to a much larger extent than high-frequency noise.

(b) The network is instantly able to fit the low frequency noise signal, causing the validation performance (w.r.t. clean validation labels without noise) to suffer. But the high-frequency noise is only fit later in the training, which first leads to a “dip” in the validation loss (indicating lower loss and better generalization) in the early stages of training, after which the loss starts to degrade. The “dip” becomes larger as the noise frequency is increased, indicating an impedance towards learning high frequency label noise in the early parts of training, when only the true (low-frequency) labels are learned. It is only as the training progresses that the higher frequencies are fit, which results in a gradually increasing (clean) validation loss.

We repeat the experiment for $\Pi$-Nets and contrast the nature of the plots w.r.t. standard networks. Since $\Pi$-Nets pick up high frequency variations faster than standard networks, we expect that the "dip" w.r.t. a fixed frequency should be smaller for $\Pi$-Nets than the standard networks i.e., we expect a higher validation loss in the presence of high frequency noise.

* For the first comparison, we consider the performance of a fully connected standard network and compare it to a $\Pi$-Net, by supplementing the fully connected network with exactly one multiplicative layer.  We verify that while the performance for the two networks is almost identical on MNIST without any label noise (corresponding to freq=0 in the legend), the validation dip for $\Pi$-Nets for higher frequencies (0.3, 0.5, 1) becomes visibly smaller (at around &lt; 100 iterations) while there is almost no dip for some frequencies (0.1, 0.2), indicating pi-networks pick up the label noise in the decision boundaries corresponding to higher frequencies faster. **([https://imgur.com/a/PPy28uH](https://imgur.com/a/PPy28uH))**, **Zoomed in for first 1000 iterations**: (**https://imgur.com/a/U9oTb7t**)

* Next, we observe the effect of increasing multiplicative injections, by comparing the first $\Pi$-Net with one multiplicative layer to the $\Pi$-Net with two multiplicative layers. As expected, the validation dip reduces even further for the network with more multiplicative layers (i.e., a higher degree polynomial) indicating that more multiplicative interactions improve the network’s ability to learn more complex decision boundaries (introduced by the high frequency noise). **(https://imgur.com/a/79HxYxZ)**, **Zoomed in for first 1000 iterations**: (**[https://imgur.com/a/jGHTQtG](https://imgur.com/a/jGHTQtG)**)

Our results allow us to make a further overarching conclusion that $\Pi$-Nets, in addition to picking up higher frequencies w.r.t inputs faster (as demonstrated in the DIP settings), in classification settings can also pick up high frequency variations in the decision boundaries faster, _thus making our general claims about the spectral bias of $\Pi$-Nets and multiplicative interactions stronger_. This also yields a further research direction towards analyzing the smoothness of decision boundaries of $\Pi$-Nets with multiplicative interactions in classification settings, especially focusing on areas where this effect becomes relevant such as adversarial susceptibility or knowledge distillation.

References

[A] On the Spectral Bias of Neural Networks: Rahaman et al., ICML 2019

---

> ### Comment · Reviewer_KnZp · 2021-11-29
> **Followup comments**
>
> Thank you for including this extra experiment; I appreciate the inclusion of results on image classification, and I think it definitely strengthens the results (and would like to see it in the paper if the authors can find space). That said, the figures could be cleaned up a little; it is difficult to see what is going on, particularly in the second set of figures, because most of the "action" happens early in training, and many curves are overlapping each other. Perhaps the authors could zoom in on the first 1000 epochs or so, and include a smaller set of frequencies if the curves are still difficult to disentangle at that scale.

---

> > ### Author Response · Authors · 2021-11-29
> > **Response to Comment from Reviewer KnZp**
> >
> > We are pleased to note that the reviewer finds the experiment valuable. As per the suggestion, we include the zoomed-in plots for the first 1000 iterations:
> >
> > * $\Pi$-Net vs Standard Net: (**https://imgur.com/a/U9oTb7t**)
> > * $\Pi$-Net (1 mult. layer) vs $\Pi$-Net (2 mult. layers): (**[https://imgur.com/a/jGHTQtG](https://imgur.com/a/jGHTQtG)**)
> >
> > The zoomed-in plots demonstrate our point more clearly and we thank the reviewer for the suggestion. *We will include the experiment in the manuscript in the camera-ready version*, especially since the reviewer has provided a reasoning as to why this is considered a valuable contribution, in addition to the DIP setting.

---

> > > ### Comment · Reviewer_KnZp · 2021-11-29
> > > **Score increase**
> > >
> > > Indeed, based on the response and additional experiment in the classification setting, I have increased my score to 8.

---

### Decision · Program_Chairs · 2022-01-20

**Decision:**

Accept (Poster)

**Comment:**

*Summary:* Investigate the NTK of PNNs and enhanced bias towards higher frequencies.

*Strengths:*
- Spectral bias is a contemporary topic.
- Some reviewers found the paper well written.

*Weaknesses:*
- Restricted setting (two-layers / no bias / infinite width), particularly in view of the objective to provide architecture design guidance. Restricted experiments (Introduction indicates learning spherical harmonics).
- Sparse discussion of related works, particularly on spectral bias.

*Discussion:*

During the discussion period authors made efforts to address some of the concerns of the reviewers. A late new experiment prompted KnZp to raise score. TQnp found the paper good but also expected a more profound theorem addressing broader PNN families given the existing work. They found that experiments and discussion of prior work could be improved. The authors added discussion of prior works and provided an explanation for their choices, but left extensions and further analysis for future work. nFMY expressed concerns about applicability of the analysis and evidence in experiments. Author responses addresses this in part. cEcf points out that the main theoretical contributions have straight forward proofs based on previous works and asks about extensions. Authors agree that the paper does not introduce novel techniques and that extending the analysis is an important direction, but leave this for future work. FuRi finds the paper provides an interesting viewpoint and raised score from 3 to 5 following the discussion (improving presentation, rigor, clarity), but considers that the paper has several drawbacks (oversimplification, lack of technical novelty) that need to be addressed.

*Conclusion:*

One reviewer found this work marginally below the acceptance threshold, three marginally above, and one good. I find that the paper considers an interesting problem and makes some interesting observations and some valuable advances. I appreciate the authors’ efforts during the reviewing period. Hence I am recommending accept. At the same time, I find that, clarity, technical and experimental contributions still can be improved and encourage the authors to carefully consider the reviewers comments when preparing the final version of the paper.